# Polyline Path Masked Attention for Vision Transformer

**Zhongchen Zhao**[1], **Chaodong Xiao**[2,3], **Hui Lin**[1], **Qi Xie**[1,*], **Lei Zhang**[2,3], **Deyu Meng**[1,4]

[1]Xi'an Jiaotong University    [2]The Hong Kong Polytechnic University    [3]OPPO Research
[4]Pazhou Laboratory (Huangpu) Institute
zhongchenzhao@stu.xjtu.edu.cn, xie.qi@mail.xjtu.edu.cn

## Abstract

Global dependency modeling and spatial position modeling are two core issues of the foundational architecture design in current deep learning frameworks. Recently, Vision Transformers (ViTs) have achieved remarkable success in computer vision, leveraging the powerful global dependency modeling capability of the self-attention mechanism. Furthermore, Mamba2 has demonstrated its significant potential in natural language processing tasks by explicitly modeling the spatial adjacency prior through the structured mask. In this paper, we propose **Polyline Path Masked Attention** (**PPMA**) that integrates the self-attention mechanism of ViTs with an enhanced structured mask of Mamba2, harnessing the complementary strengths of both architectures. Specifically, we first ameliorate the traditional structured mask of Mamba2 by introducing a 2D polyline path scanning strategy and derive its corresponding structured mask, polyline path mask, which better preserves the adjacency relationships among image tokens. Notably, we conduct a thorough theoretical analysis on the structural characteristics of the proposed polyline path mask and design an efficient algorithm for the computation of the polyline path mask. Next, we embed the polyline path mask into the self-attention mechanism of ViTs, enabling explicit modeling of spatial adjacency prior. Extensive experiments on standard benchmarks, including image classification, object detection, and segmentation, demonstrate that our model outperforms previous state-of-the-art approaches based on both state-space models and Transformers. For example, our proposed PPMA-T/S/B models achieve **48.7%/51.1%/52.3%** mIoU on the ADE20K semantic segmentation task, surpassing RMT-T/S/B by 0.7%/1.3%/0.3%, respectively. Code is available at `https://github.com/zhongchenzhao/PPMA`.

## 1 Introduction

The research of foundational models has long been a cornerstone of deep learning. In computer vision, Convolutional Neural Networks (CNNs) [20, 19] and Vision Transformers (ViTs) [46, 9, 31, 22] currently represent the dominant architectures. Notably, ViTs have become the most mainstream architecture in large models through the powerful self-attention mechanism, which can capture the non-local self-similarity within global receptive fields. However, the quadratic complexity of the Transformer, when implementing self-attention, severely limits its application in large image processing models. Moreover, as shown in Fig. 1 (b), classic positional encoding methods [46, 31, 38] in ViTs lack the explicit modeling capability of spatial distance between image tokens, and largely ignore the important spatial adjacency priors in texture, shape, semantics, and so on. This increases the learning pressure and limits its capability for fine-grained image feature extraction.

Compared to CNNs and Transformers, the recently proposed Mamba [11] achieves linear complexity while maintaining global receptive fields, demonstrating strong potential as the next-generation architecture. Specifically, Mamba follows the State Space Models (SSMs) paradigm and employs the selective scan mechanism with the state transition matrix to recursively propagate dependencies among

---

*Corresponding author.

39th Conference on Neural Information Processing Systems (NeurIPS 2025).

tokens in a sequence. Building on this foundation, Mamba2 [6] further refines the state transition matrix into a lightweight structured mask and introduces a unified theoretical framework, Structured State Space Duality (SSD), to bridge SSMs and attention variants. Under SSD, the core selective scan mechanism of Mamba2 can be reformulated as a form of structured masked attention, *i.e.*, a **Linear Attention** [23] element-wise multiplied by the **structured mask**, as illustrated in Fig. 1 (a). Notably, this structured mask explicitly encodes the sequence adjacency of tokens,

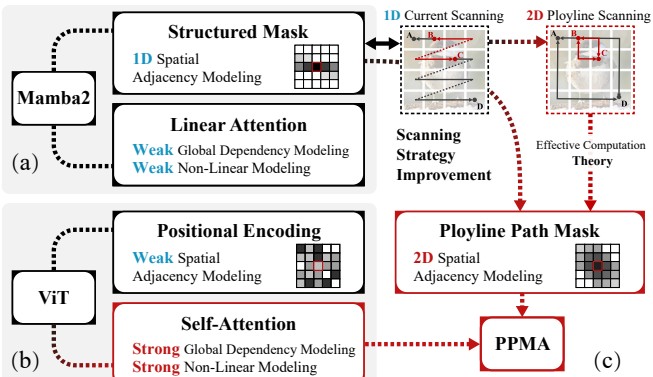

Figure 1: (a)-(b) Illustration of the modules in Mamba2 and ViT. (c) Our method adapts the structured mask of Mamba2 to 2D scanning and integrates it with ViT's self-attention.

enabling Mamba2 to match or surpass Transformers across various natural language processing (NLP) tasks. Following its success in NLP, Mamba [11] has been rapidly adapted to various visual domains, including: high-level tasks (classification, object detection, segmentation [58, 30, 49, 47]), low-level tasks (super-resolution, denoising, deraining [13, 12, 59]), image generation [43], video analysis [26], point cloud analysis [27, 51], and remote sensing images [54].

Although Mamba [11] has demonstrated impressive results on certain vision tasks, empirical results on high-level vision tasks demonstrate that even state-of-the-art (SOTA) Mamba-based backbones [30, 49, 47, 15] still underperform SOTA Transformer-based backbones [57, 10] with a substantial performance gap. As shown in Fig. 1 (a), this gap mainly stems from two issues: (I) **1D Scanning Issue.** Mamba's 1D scanning strategy arranges the tokens of a 2D image into a 1D sequence, which inevitably disrupts the inherent spatial adjacency within 2D images and limits the effectiveness of its recursive selective scanning mechanism. (II) **Weak Global Dependency Modeling Issue.** The linear attention in Mamba2 omits the non-linear softmax layer, leading to a decrease in the precision and stability of global dependency modeling of images.

In this paper, we present **Polyline Path Masked Attention** (**PPMA**), a brand-new method that effectively combines the advantages of ViTs and Mamba2. Specifically, as illustrated in Fig. 1 (c), to address the 1D Scanning Issue when applying current Mamba to 2D images, we propose a novel 2D polyline path scanning strategy and derive an efficient calculation method for its corresponding structured mask, the polyline path mask. Then, we embed the polyline path mask as an explicit positional encoding into the ViT framework. This not only avoids the Weak Global Dependency Modeling Issue of Mamba2, but also alleviates the positional encoding issue of ViTs. As a result, our method fully leverages the powerful global context modeling capability of the self-attention mechanism in ViTs together with the explicit spatial adjacency modeling capability of the polyline path mask inspired by Mamba2, achieving SOTA performance on mainstream high-level vision tasks.

To the best of our knowledge, this is the first work to integrate Mamba2's structured mask mechanism into ViTs. The main contributions of this study are summarized as follows:

- We propose a 2D polyline path scanning strategy for visual Mamba, which better preserves the inherent 2D spatial structure of images compared to existing scanning strategies. Building on this, we further derive a novel structured mask, termed polyline path mask, which is more suitable for 2D images than the traditional structured mask used in Mamba2.

- We conduct a comprehensive theoretical analysis for the proposed 2D polyline path mask. Specifically, we theoretically prove that it can be decomposed into two 1D structured masks with clear physical meanings (i.e., horizontal and vertical scanning masks). More importantly, by leveraging this decomposability, we derive an efficient algorithm to reduce its computational complexity from $\mathcal{O}(N^2)$ in the naive calculation to $\mathcal{O}(N^{\frac{3}{2}})$.

- The polyline path mask can be seamlessly integrated into various attention variants in a plug-and-play manner without introducing a substantial increase in computational complexity. In this paper, we incorporate it into the vanilla self-attention and criss-cross attention, deriving the Polyline Path Masked Attention (PPMA).

- Leveraging PPMA, we construct a hybrid Mamba2-Transformer model. Experimental results demonstrate that our model achieves SOTA performance on standard benchmarks for image classification, object detection, and segmentation.

## 2 Related Work

**Vision Transformers.** ViTs have become foundational in large-scale vision models such as SAM [24] and Sora [1], primarily due to their self-attention mechanism that effectively captures the long-range dependency. Moreover, the spatial structural information provided by positional encodings (e.g., APE [46], RPE [31], and RoPE [38]) is also crucial to ViTs. However, traditional positional encodings fail to explicitly encode spatial adjacency. Recent works, such as RMT [10] and VVT [39], porpose to incorporate RetNet's input-independent temporal decay mask [40] into ViTs for more explicit spatial modeling based on the Manhattan distance. In comparison, Mamba2's input-dependent selective structured mask not only explicitly encodes the relative positional information in the spatial space but also captures the semantic continuity in the feature space.

**Mamba.** As a state space model, Mamba introduces an input-dependent selection mechanism into the state transition matrix $\mathbf{A}$, achieving Transformer-level performance with linear complexity on NLP tasks. Building on this foundation, Mamba2 [6] further simplifies the matrix $\mathbf{A}$ to a scalar $a$, enabling more hardware-efficient parallelizable training without sacrificing performance. Moreover, Mamba2 [6] demonstrates that its formulation is mathematically equivalent to a 1-semiseparable structured masked attention, and develops the State Space Duality (SSD) framework to connect structured SSMs and attention variants. Furthermore, Mamba2 points out that other potential structured masked attentions can also be integrated into the SSD framework.

In this paper, we introduce a novel structured masked attention, termed polyline path masked attention, tailored for vision tasks. Different from the previous 2D selective SSM framework [53] based on Mamba [11], our Mamba2-based polyline path mask is more lightweight and can be plugged seamlessly into various attention variants. Moreover, compared to MambaVision [17] which naively concatenates Mamba's blocks and ViT self-attention layers, our method more effectively harnesses complementary strengths of both architectures.

## 3 Preliminaries

**Mamba2's Recurrent Form.** Mamba2 [6] initially adopts a recurrent form with linear complexity for sequence modeling. Specifically, Mamba2 employs the selective state space models to map the input sequence $\boldsymbol{x} \in \mathbb{R}^{N \times C}$ to the output sequence $\boldsymbol{y} \in \mathbb{R}^{N \times C}$, i.e., for $i = 1 : N$,

$$\boldsymbol{h}_i = a_i \boldsymbol{h}_{i-1} + \boldsymbol{B}_i^\top \boldsymbol{x}_i, \qquad \boldsymbol{y}_i = \boldsymbol{C}_i \boldsymbol{h}_i, \tag{1}$$

where $\boldsymbol{x}_i, \boldsymbol{y}_i \in \mathbb{R}^{1 \times C}$, $\boldsymbol{h}_i \in \mathbb{R}^{D \times C}$ denotes the hidden state, $a_i \in \mathbb{R}$ and $\boldsymbol{B}_i, \boldsymbol{C}_i \in \mathbb{R}^{1 \times D}$ are input-dependent parameters learned by multilayer perceptron (MLP) layers, the scalar $a_i$ serves as a decay factor bounded in $[0, 1]$, $N, C$ and $D$ denote the sequence length, channel number, and hidden state dimension, respectively.

**Mamba2's Attention Form.** Leveraging the SSD framework in Mamba2 [6], the recurrent form of Mamba2 in Eq. (1) can be reformulated as its equivalent dual form, i.e., structured masked attention, by eliminating the hidden state $\boldsymbol{h_i}$ via substitution:

$$\boldsymbol{y} = \left( \boldsymbol{C}\boldsymbol{B}^\top \odot \boldsymbol{L}^{1D} \right) \boldsymbol{x}, \qquad \boldsymbol{L}_{ij}^{1D} = a_{i:j} = \begin{cases} a_i \times \cdots \times a_{j+1} & i > j \\ 1 & i = j \ , \\ 0 & i < j \end{cases} \tag{2}$$

where $\odot$ denotes the Hadamard (element-wise) product, $\boldsymbol{B}, \boldsymbol{C} \in \mathbb{R}^{N \times D}$, and the 1D structured mask $\boldsymbol{L}^{1D} \in \mathbb{R}^{N \times N}$ is a 1-semiseparable matrix which can be efficiently calculated with a complexity of $\mathcal{O}(N^2)$ by the chunkwise algorithm [6]. Mamba2's attention form (Eq. (2)) enables more efficient parallelizable training than its recurrent form (Eq. (1)). Notably, parameters $\boldsymbol{C}$ and $\boldsymbol{B}$ in Eq. (2) are learned analogously to the query $\boldsymbol{Q}$ and key $\boldsymbol{K}$ in ViTs, respectively. Thus, Eq. (2) reveals that the selective state transition function in Mamba2 is equivalent to the Hadamard product of a linear attention map $\boldsymbol{C}\boldsymbol{B}^\top$ and a 1D structured mask $\boldsymbol{L}^{1D}$. Here, the structured mask can be interpreted as a form of relative positional encoding [6].

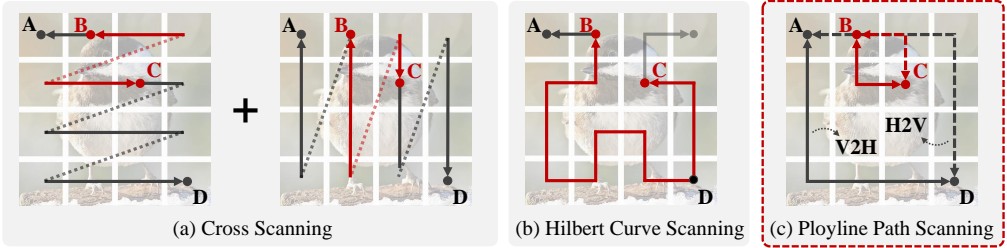

(a) Cross Scanning      (b) Hilbert Curve Scanning    (c) Ployline Path Scanning

Figure 2: Compared to existing scanning strategies (a) and (b), which flatten 2D tokens into a 1D sequence, our polyline path scanning (c) better preserves the adjacency of 2D tokens.

In this work, we extend the structured masked attention in Mamba2 from 1D sequences to 2D images. Specifically, we extend the 1D structured mask $L^{1D}$ to the 2D polyline path mask $L^{2D}$ by introducing a novel 2D scanning strategy, and propose an efficient algorithm for computing and applying this polyline path mask $L^{2D}$. The proposed $L^{2D}$ can be substituted into Eq. (2) for replacing $L^{1D}$ or adopted in ViTs as the explicit positional encoding.

# 4 Method

In this section, we introduce the idea of adapting the structured mask of Mamba2 to 2D scanning and integrating it into the self-attention mechanism of ViTs, achieving an explicit positional encoding. Specifically, we 1) introduce the definition of 2D polyline path mask in Sec. 4.1; 2) analyze the theoretical properties of the proposed polyline path mask and introduce an efficient algorithm for the proposed polyline path mask in Sec. 4.2; 3) apply the polyline path mask to standard self-attention and criss-cross attention of ViTs in Sec. 4.3.

## 4.1 Definition of Polyline Path Mask

As a sequence autoregressive framework, visual Mamba starts from employing a scanning strategy to flatten a 2D image into a 1D sequence of image tokens. This scanning strategy plays an important role in Mamba's performance, since the order of tokens is determined by it. As illustrated in Fig. 2, previous works [58, 30, 27] have proposed various scanning strategies for visual Mamba. However, these strategies fail to fully preserve the inherent spatial adjacency of 2D tokens. For example, as shown in Fig. 2 (a) and (b), for two tokens $B$ and $C$ which are close in an image, previous scanning strategies [30, 27] may cause them to be significantly farther apart in the 1D scanning path.

**Polyline Path Scanning.** To address this limitation, we design a 2D polyline path scanning strategy. Specifically, for each token pair $(\boldsymbol{x}_{i,j}, \boldsymbol{x}_{k,l})$ in the 2D grid, we define their scanning path as the L-shaped polyline connecting them, as shown in Fig. 2 (c). To ensure symmetry in mutual distances, we set two bidirectional polyline paths: vertical-then-horizontal path (V2H solid lines in Fig. 2 (c)) and horizontal-then-vertical path (H2V dotted lines in Fig. 2 (c)), and use their combination as the final scanning path. In this way, the adjacency relationship of 2D tokens can be strictly maintained under the Manhattan distance[2]. Intuitively speaking, tokens close (or far) to each other will be in close (or far) distance on the scanning path, and vice versa. As the example shown in Fig. 2, polyline scanning strategy better preserves the distance between token $B$ and $C$ compared to the other two strategies. An more intuitive example is shown in Fig. 8.

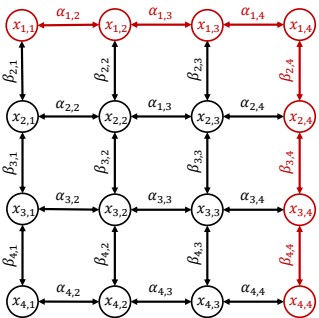

Figure 3: An intuitive example illustrating the polyline path mask on a $4 \times 4$ grid.

**Definition of Polyline Path Mask.** Based on the proposed polyline path scanning strategy, we introduce the polyline path mask. As an example shown in Fig. 3, we define the horizontal and vertical decay factors of each input token $\boldsymbol{x}_{i,j}$ as $\alpha_{i,j}$ and $\beta_{i,j}$, respectively. In this paper, we employ two MLP layers to learn $\alpha_{i,j}$ and $\beta_{i,j}$, respectively.[3] Then, the decay weight of V2H polyline path

---

[2]The Manhattan distance between two points $\boldsymbol{x}_{i,j}$ and $\boldsymbol{x}_{k,l}$ in a 2D plane is $|i - k| + |j - l|$.

[3]We apply the ReLU and exponential operator after the MLP layers to ensure $\alpha_{i,j}, \beta_{i,j} \in [0, 1]$.

from $\boldsymbol{x}_{k,l}$ to $\boldsymbol{x}_{i,j}$ is defined as $\mathcal{L}_{i,j,k,l}$ , which is the product of all decay factors along that path, i.e.,

$$\mathcal{L}_{i,j,k,l} = \alpha_{i,j:l}\beta_{i:k,l}, \text{ where } \alpha_{i,j:l} = \begin{cases} \prod_{n=j+1}^{l} \alpha_{i,n} & j < l \\ 1 & j = l \\ \prod_{n=l+1}^{j} \alpha_{i,n} & j > l \end{cases}, \ \beta_{i:k,l} = \begin{cases} \prod_{n=i+1}^{k} \beta_{n,l} & i < k \\ 1 & i = k \\ \prod_{n=k+1}^{i} \beta_{n,l} & i > k \end{cases}. \quad (3)$$

For example, as illustrated in Fig. 3, the V2H polyline path's decay weight from token $\boldsymbol{x}_{4,4}$ to $\boldsymbol{x}_{1,1}$ is $\mathcal{L}_{1,1,4,4} = \alpha_{1,2}\alpha_{1,3}\alpha_{1,4}\beta_{2,4}\beta_{3,4}\beta_{4,4}$. Similarly, the decay weight along the H2V polyline path is defined as $\tilde{\mathcal{L}}_{i,j,k,l} = \alpha_{k,j:l}\beta_{i:k,l}$. Due to the spatial symmetry, it is evident that $\tilde{\mathcal{L}}_{i,j,k,l} = \mathcal{L}_{k,l,i,j}$. By combining the V2H and H2V polyline paths, the final decay weight is

$$\mathcal{L}^{2D} = \mathcal{L} + \tilde{\mathcal{L}}. \quad (4)$$

Note that $\mathcal{L}$, $\tilde{\mathcal{L}}$ and $\mathcal{L}^{2D}$ are all 4D tensors of size $\mathbb{R}^{H \times W \times H \times W}$, where $H$ and $W$ are the height and width of the feature map, respectively. The polyline path mask, a 2D matrix $\boldsymbol{L}^{2D} \in \mathbb{R}^{HW \times HW}$, can be obtained by unfolding the decay weight tensor, i.e., for all $i$, $j$, $k$, and $l$,

$$\left[\boldsymbol{L}^{2D}\right]_{(i-1)\times W+j, (k-1)\times W+l} = \mathcal{L}^{2D}_{i,j,k,l}. \quad (5)$$

For simplicity, we denote the above tensor-to-matrix unfolding operation as $\boldsymbol{L}^{2D} = \text{unfold}(\mathcal{L}^{2D})$, and its inverse operation as $\mathcal{L}^{2D} = \text{fold}(\boldsymbol{L}^{2D})$ in the following sections. More details can be found in Appendix A.2.

## 4.2 Efficient Computation Theory of Polyline Path Mask

According to the definition (3), the direct approach to compute the polyline path mask $\boldsymbol{L}^{2D}$ is to calculate each element individually. However, the large size of the mask and numerous multiplications for each element lead to a high computational cost in both calculating and applying $\boldsymbol{L}^{2D}$. To address this issue, we present a decomposition theorem for matrices structured as $\boldsymbol{L}^{2D}$. Based on this, we further design an efficient algorithm for performing multiplication on $\boldsymbol{L}^{2D}$. For simplicity, we focus our theoretical study on $\boldsymbol{L}$, which is similar to the case of $\boldsymbol{L}^{2D}$. Complete proofs of the theorems are provided in Appendix A.3 and A.5.

**Theorem 1** (Matrix Decomposition). *For any matrix $\boldsymbol{M} \in \mathbb{R}^{HW \times HW}$ and $\mathcal{M} = \text{fold}(\boldsymbol{M})$, if for $\forall i, j, k, l$, $\exists \boldsymbol{A}^i \in \mathbb{R}^{W \times W}$ and $\boldsymbol{B}^l \in \mathbb{R}^{H \times H}$, s.t., $\mathcal{M}_{i,j,k,l} = \left[\boldsymbol{A}^i\right]_{j,l} \times \left[\boldsymbol{B}^l\right]_{i,k}$, then $\boldsymbol{M}$ can be decomposed as:*

$$\boldsymbol{M} = \boldsymbol{M}^A \times \boldsymbol{M}^B = \hat{\boldsymbol{M}}^A \odot \hat{\boldsymbol{M}}^B, \quad (6)$$

*where $\boldsymbol{M}^A, \boldsymbol{M}^B, \hat{\boldsymbol{M}}^A, \hat{\boldsymbol{M}}^B \in \mathbb{R}^{HW \times HW}$, which satisfy*

$$\boldsymbol{M}^A = \text{unfold}(\mathcal{M}^A), \boldsymbol{M}^B = \text{unfold}(\mathcal{M}^B), \text{ s.t., } \mathcal{M}^A_{i,:,k,:} = \begin{cases} \boldsymbol{A}^i & k=i \\ 0 & k \neq i \end{cases}, \ \mathcal{M}^B_{:,j,:,l} = \begin{cases} \boldsymbol{B}^l & j=l \\ 0 & j \neq l \end{cases}, \quad (7)$$

$$\hat{\boldsymbol{M}}^A = \text{unfold}(\hat{\mathcal{M}}^A), \ \hat{\boldsymbol{M}}^B = \text{unfold}(\hat{\mathcal{M}}^B), \ \text{ s.t., } \ \hat{\mathcal{M}}^A_{i,:,k,:} = \boldsymbol{A}^i, \ \hat{\mathcal{M}}^B_{:,j,:,l} = \boldsymbol{B}^l. \quad (8)$$

As defined in Eq. (3), the polyline path mask $\boldsymbol{L}$ satisfies the conditions in Theorem 1 with $[\boldsymbol{A}^i]_{j,l} = \alpha_{i,j:l}$ and $[\boldsymbol{B}^l]_{i,k} = \beta_{i:k,l}$. Thus, based on Theorem 1, the polyline path mask $\boldsymbol{L}$ can be decomposed as $\boldsymbol{L} = \boldsymbol{L}^H \times \boldsymbol{L}^V = \hat{\boldsymbol{L}}^H \odot \hat{\boldsymbol{L}}^V$. Moreover, for the complexity of computing $\boldsymbol{L}$, we have:

**Corollary 1** (Mask Complexity). *The complexity of directly computing polyline path mask $\boldsymbol{L}$ with Eq.(3) and (5) is $\mathcal{O}(N^{\frac{5}{2}})$, which can be reduced to $\mathcal{O}(N^2)$ by applying Theorem 1, where $N = H \times W$.*

For matrices in the form of Eq. (7), when performing multiplication operations, we have:

**Theorem 2** (Efficient Matrix Multiplication). *For matrices $\boldsymbol{M}^A, \boldsymbol{M}^B$ defined in Eq. (7), $\forall \boldsymbol{x} \in \mathbb{R}^{HW}$, the following equation holds:*

$$\boldsymbol{y} = \boldsymbol{M}^A \times \boldsymbol{M}^B \times \boldsymbol{x} \quad \Leftrightarrow \quad \boldsymbol{Z}_{:,l} = \boldsymbol{B}^l \times \boldsymbol{X}_{:,l}, \ \boldsymbol{Y}_{i,:} = \boldsymbol{A}^i \times \boldsymbol{Z}_{i,:}, \quad (9)$$

*where $\boldsymbol{y} \in \mathbb{R}^{HW}$, $\boldsymbol{X} = \text{unvec}(\boldsymbol{x}) \in \mathbb{R}^{H \times W}$, $\boldsymbol{Y} = \text{unvec}(\boldsymbol{y}) \in \mathbb{R}^{H \times W}$, $\boldsymbol{Z} \in \mathbb{R}^{H \times W}$, and the operator $\text{vec}(\cdot)$ vectorizes a matrix by stacking its columns and $\text{unvec}(\cdot)$ is its inverse operator.*

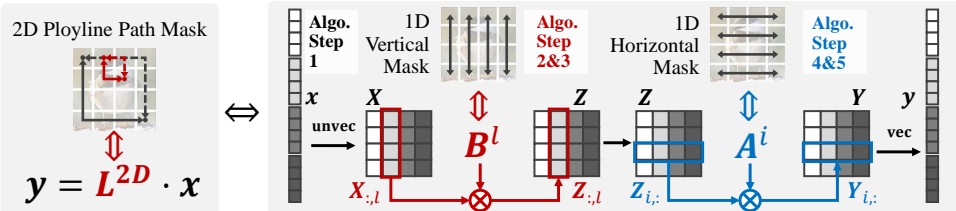

Figure 4: Illustration of the efficient algorithm for utilizing the proposed polyline path mask. Left: Naive computation of matrix multiplication. Right: An intuitive illustration of Algorithm 1.

Based on Theorem 2, we can design Algorithm 1 for computing the matrix multiplication between polyline path mask $\boldsymbol{L}$ and the vector $\boldsymbol{x}$. Note that the involved matrices $\boldsymbol{A}^i$ and $\boldsymbol{B}^l$ are symmetric matrices with lower triangular parts being 1-semiseparable, as defined in Mamba2 [6]. This will lead to a substantial reduction in complexity as stated in the following corollary.

---

**Algorithm 1: Efficient Masked Attention Computation.**

---

**Input:** decay factors $\alpha, \beta$ of $\boldsymbol{L}$, vector $\boldsymbol{x} \in \mathbb{R}^{HW}$;
1: Compute $\boldsymbol{X} = \mathrm{unvec}(\boldsymbol{x}) \in \mathbb{R}^{H \times W}$;
2: Compute $\boldsymbol{B}^l \in \mathbb{R}^{H \times H}$, where for $l = 1:W$, $[\boldsymbol{B}^l]_{i,k} = \beta_{i:k,l}$;
3: Compute $\boldsymbol{Z} \in \mathbb{R}^{H \times W}$, where $\boldsymbol{Z}_{:,l} = \boldsymbol{B}^l \times \boldsymbol{X}_{:,l}$;
4: Compute $\boldsymbol{A}^i \in \mathbb{R}^{W \times W}$, where for $i = 1:H$, $[\boldsymbol{A}^i]_{j,l} = \alpha_{i,j:l}$;
5: Compute $\boldsymbol{Y} \in \mathbb{R}^{H \times W}$, where $\boldsymbol{Y}_{i,:} = \boldsymbol{A}^i \times \boldsymbol{Z}_{i,:}$;
**Output:** $\boldsymbol{y} = \mathrm{vec}(\boldsymbol{Y})$;

---

**Corollary 2** (Masked Attention Complexity). *The computational complexity of the matrix multiplication between polyline path mask and vector $\boldsymbol{x}$, i.e., $\boldsymbol{y} = \boldsymbol{L}\boldsymbol{x}$, can be reduced from $\mathcal{O}(N^2)$ to $\mathcal{O}(N^{\frac{3}{2}})$ by Algorithm 1, and further reduced to $\mathcal{O}(N)$ by applying the chunkwise algorithm of Mamba2 [6] to steps 3 and 5 in Algorithm 1.*

**Remarks.** Intuitively, as illustrated in Fig. 4, Algorithm 1 shows that the 2D polyline path scanning on 2D tokens (i.e., $\boldsymbol{L}\boldsymbol{x}$) can be decomposed as the 1D vertical scanning along each column of $\boldsymbol{X}$ (i.e., $\boldsymbol{Z}_{:,l} = \boldsymbol{B}^l \times \boldsymbol{X}_{:,l}$) followed by the 1D horizontal scanning along each row of $\boldsymbol{Z}$ (i.e., $\boldsymbol{Y}_{i,:} = \boldsymbol{A}^i \times \boldsymbol{Z}_{i,:}$). This equivalence offers an intuitive understanding of the physical meaning of the decomposed polyline path mask $\boldsymbol{L} = \boldsymbol{L}^H \boldsymbol{L}^V$ and enables its natural extension to 3D or higher-dimensional tokens, as detailed in Appendix C.2.

## 4.3 Polyline Path Masked Attention

The proposed polyline path mask can be seamlessly integrated into various attention variants in a plug-and-play manner. In this section, we integrate it into two softmax-based self-attention layers: vanilla attention [9] and criss-cross attention [22]. Notably, theorems and algorithm given in Sec. 4.2 guarantee that integration of polyline path mask does not substantially increase the computational complexity of the original attention mechanism. More applications, such as the polyline path masked linear attention with a complexity of $\mathcal{O}(N)$, are provided in Appendix A.7.

**Polyline Path Masked Vanilla Attention.** The polyline path mask $\boldsymbol{L}^{2D}$ is integrated into vanilla attention via a Hadamard product with the attention map, i.e., for query $\boldsymbol{Q}$, key $\boldsymbol{K}$, and value $\boldsymbol{V}$:

$$\mathrm{PPMVA}\left(\boldsymbol{x}\right) = \left(\mathrm{softmax}(\boldsymbol{Q}\boldsymbol{K}^{\top}) \odot \boldsymbol{L}^{2D}\right) \boldsymbol{V}, \tag{10}$$

where $\boldsymbol{Q}, \boldsymbol{K}, \boldsymbol{V} \in \mathbb{R}^{HW \times C}$. Based on Corollary 1, Eq. (10) maintains the complexity of $\mathcal{O}(N^2)$.

**Polyline Path Masked Criss-Cross Attention.** The original criss-cross attention [22] employs the sparse attention over tokens located in the same row or column, achieving a complexity of $\mathcal{O}(N^{\frac{3}{2}})$. In this work, we follow RMT [10] to decompose criss-cross attention into the vertical attention over each column followed by the horizontal attention over each row. The polyline path mask $\boldsymbol{L}^{2D}$ is applied to the decomposed criss-cross attention through the Hadamard product, that is:

$$\mathrm{PPMCCA}\left(\boldsymbol{x}\right) = \left(\left(\boldsymbol{S}^H \times \boldsymbol{S}^V\right) \odot \boldsymbol{L}^{2D}\right)\boldsymbol{V} = \left(\left(\boldsymbol{S}^H \times \boldsymbol{S}^V\right) \odot \boldsymbol{L}\right)\boldsymbol{V} + \left(\left(\boldsymbol{S}^H \times \boldsymbol{S}^V\right) \odot \tilde{\boldsymbol{L}}\right)\boldsymbol{V}, \tag{11}$$

where horizontal and vertical attention maps $\boldsymbol{S}^H, \boldsymbol{S}^V \in \mathbb{R}^{HW \times HW}$ satisfy the form in Eq. (7) with $\boldsymbol{A}^i = \mathrm{softmax}(\boldsymbol{\mathcal{Q}}_{i,:,:}\boldsymbol{\mathcal{K}}_{i,:,:}^{\top})$ and $\boldsymbol{B}^l = \mathrm{softmax}(\boldsymbol{\mathcal{Q}}_{:,l,:}\boldsymbol{\mathcal{K}}_{:,l,:}^{\top})$, and $\boldsymbol{\mathcal{Q}}, \boldsymbol{\mathcal{K}} \in \mathbb{R}^{H \times W \times C}$ are tensor forms

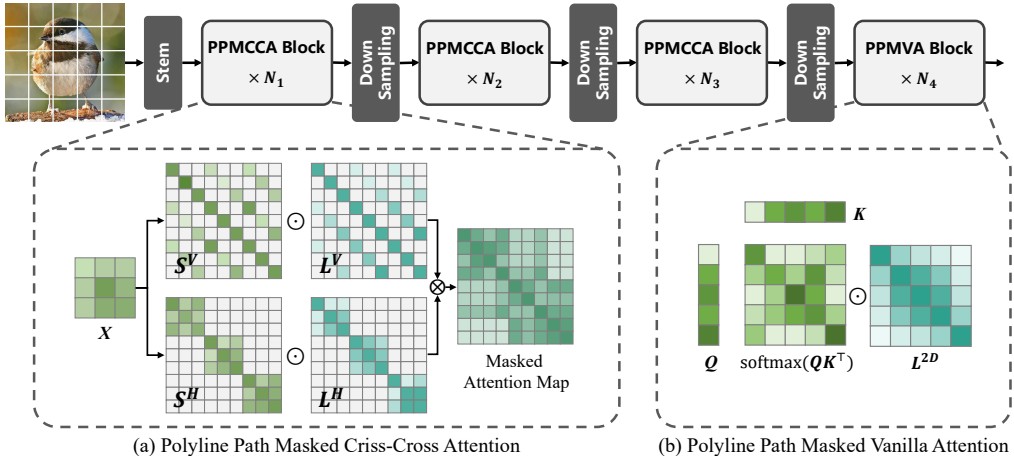

(a) Polyline Path Masked Criss-Cross Attention      (b) Polyline Path Masked Vanilla Attention

Figure 5: Overall architecture of the Polyline Path Masked Attention based Vision Transformer.

of $Q, K$, respectively [22]. Based on Theorem 1, we can reformulate the left part of Eq. (11) as:

$$
\left(\left(S^H \times S^V\right) \odot L\right)V = \left(\left(S^H \times S^V\right) \odot \left(L^H \times L^V\right)\right)V = \left(\left(\hat{S}^H \odot \hat{S}^V\right) \odot \left(\hat{L}^H \odot \hat{L}^V\right)\right)V
$$
$$
= \left(\left(\hat{S}^H \odot \hat{L}^H\right) \odot \left(\hat{S}^V \odot \hat{L}^V\right)\right)V = \left(S^H \odot L^H\right) \times \left(S^V \odot L^V\right) \times V. \tag{12}
$$

Note that matrices $\hat{S}^H \odot \hat{L}^H$ and $\hat{S}^V \odot \hat{L}^V$ also satisfy the form in Eq. (7). Thus, the computational complexity of Eq. (12) can be reduced to $\mathcal{O}(N^{\frac{3}{2}})$ by Algorithm 1. Similar conclusions can also be derived for the right part of Eq. (11). Thus, the complexity of Eq.(11) maintains $\mathcal{O}(N^{\frac{3}{2}})$.

### 4.4 Overall Architecture

Based on the proposed Polyline Path Masked Attention, we construct a hybrid Mamba2-Transformer backbone for vision tasks. As illustrated in Fig. 5, our backbone adopts the four-stage hierarchical architecture. Following RMT [10], we employ Polyline Path Masked Criss-Cross Attention in the first three stages, and Polyline Path Masked Vanilla Attention in the final stage. Moreover, we develop our model in three scales: tiny (PPMA-T), small (PPMA-S), and base (PPMA-B).

## 5 Experiments

To validate the effectiveness of our method, we conduct a series of experiments on mainstream benchmarks for image classification (Sec. 5.1), object detection and instance segmentation (Sec. 5.2), and semantic segmentation (Sec. 5.3). Comparison methods include advanced CNN-based [34, 32, 42], SSM-based [30, 53, 49, 47], and Transformer-based backbones [31, 8, 17, 16, 57, 10]. For a fair comparison, we reproduce the experimental results of RMT [10] with the same experimental settings as ours. We also perform comprehensive ablation studies on the structured mask design in Sec. 5.4. More detailed experimental settings and results can be found in Appendix B.

### 5.1 Image Classification on ImageNet-1K

**Settings.** We evaluate the classification performance of our method on ImageNet-1K [7]. Following the same training strategy as in [10, 44], we train our models from scratch for 300 epochs with the input size of 224×224. We use the adaptive AdamW optimizer with a cosine decay learning rate scheduler (batch size=1024, initial learning rate=0.001, weight decay=0.05).

**Results.** The comparison results presented in Table 1 show that our method achieves state-of-the-art (SOTA) performance compared to other advanced models based on various architectures across tiny, small, and base scales. Specifically, PPMA-S achieves **84.2%** top-1 accuracy, surpassing 2DMamba-T [53] by **1.4%**, MLLA-T [15] by **0.7%**, MambaVision-T2 [17] by **1.5%**, and RMT-S [10] by **0.2%** with similar FLOPs. PPMA-T achieves **82.6%** top-1 accuracy, outperforming the most competitive RMT-T [10] by **0.2%** without extra training tricks. Moreover, our PPMA-B also surpasses other SOTA CNN-based, SSM-based, and Transformer-based backbones.

Table 1: Image classification performance on the ImageNet-1K validation set.

| Model | Arch. | #Param. (M) | FLOPs (G) | Top-1 (%) |
|---|---|---|---|---|
| RegNetY-1.6G [34] | CNN | 11 | 1.6 | 78.0 |
| EffNet-B3 [42] | CNN | 12 | 1.8 | 81.6 |
| Vim-T [58] | SSM | 7 | 1.5 | 76.1 |
| MSVMamba-M [36] | SSM | 12 | 1.5 | 79.8 |
| BiFormer-T [57] | Trans. | 13 | 2.2 | 81.4 |
| NAT-M [16] | Trans. | 20 | 2.7 | 81.8 |
| SMT-T [29] | Trans. | 12 | 2.4 | 82.2 |
| RMT-T [10] | Trans. | 14 | 2.5 | 82.4 |
| PPMA-T | | 14 | 2.7 | **82.6** |
| RegNetY-4G [34] | CNN | 21 | 4.0 | 80.0 |
| ConvNeXt-T [32] | CNN | 29 | 4.5 | 82.1 |
| EffNet-B4 [42] | CNN | 19 | 4.2 | 82.9 |
| VMamba-T [30] | SSM | 30 | 4.9 | 82.6 |
| 2DMamba-T [53] | SSM | 31 | 4.9 | 82.8 |
| GrootVL-T [49] | SSM | 30 | 4.8 | 83.4 |
| Spatial-Mamba-T [47] | SSM | 27 | 4.5 | 83.5 |
| MLLA-T [15] | SSM | 25 | 4.2 | 83.5 |
| Swin-T [31] | Trans. | 29 | 4.5 | 82.1 |
| CSWin-T [8] | Trans. | 23 | 4.3 | 82.7 |
| MambaVision-T2 [17] | Trans. | 35 | 5.1 | 82.7 |

| Model | Arch. | #Param. (M) | FLOPs (G) | Top-1 (%) |
|---|---|---|---|---|
| NAT-T [16] | Trans. | 28 | 4.3 | 83.2 |
| BiFormer-S [57] | Trans. | 26 | 4.5 | 83.8 |
| RMT-S [10] | Trans. | 27 | 4.5 | 84.0 |
| PPMA-S | | 27 | 4.9 | **84.2** |
| RegNetY-8G [34] | CNN | 39 | 8.0 | 81.7 |
| ConvNeXt-S [32] | CNN | 50 | 8.7 | 83.1 |
| EffNet-B5 [42] | CNN | 30 | 9.9 | 83.6 |
| VMamba-S [30] | SSM | 50 | 8.7 | 83.6 |
| 2DMamba-S [53] | SSM | 50 | 8.8 | 83.8 |
| GrootVL-S [49] | SSM | 51 | 8.5 | 84.2 |
| MLLA-S [15] | SSM | 43 | 7.3 | 84.4 |
| Spatial-Mamba-S [47] | SSM | 43 | 7.1 | 84.6 |
| Swin-S [31] | Trans. | 50 | 8.7 | 83.0 |
| NAT-S [16] | Trans. | 51 | 7.8 | 83.7 |
| CSWin-B [8] | Trans. | 78 | 15.0 | 84.2 |
| MambaVision-B [17] | Trans. | 98 | 15.0 | 84.2 |
| BiFormer-B [57] | Trans. | 57 | 9.8 | 84.3 |
| iFormer-B [37] | Trans. | 48 | 9.4 | 84.6 |
| RMT-B [10] | Trans. | 54 | 9.7 | 84.9 |
| PPMA-B | | 54 | 10.6 | **85.0** |

Table 2: Object detection and instance segmentation performance with Mask R-CNN [18] detector on COCO val2017. FLOPs are calculated with input resolution of $1280 \times 800$.

| Backbone | #Param. (M) | FLOPs (G) | $AP^b$ | $AP^b_{50}$ | $AP^b_{75}$ | $AP^m$ | $AP^m_{50}$ | $AP^m_{75}$ |
|---|---|---|---|---|---|---|---|---|
| **Mask R-CNN 1× schedule** | | | | | | | | |
| Vim-T [58] | – | – | 45.7 | 63.9 | 49.6 | 39.2 | 60.9 | 41.7 |
| MSVMamba-M [36] | 32 | 201 | 43.8 | 65.8 | 47.7 | 39.9 | 62.9 | 42.9 |
| MPViT-XS [25] | 30 | 231 | 44.2 | 66.7 | 48.4 | 40.4 | 63.4 | 43.4 |
| RMT-T [10] | 33 | 218 | 46.7 | 68.6 | 51.6 | 42.1 | 65.3 | 45.2 |
| PPMA-T | 33 | 218 | **47.1** | **68.7** | **51.7** | **42.4** | **65.9** | **45.7** |
| ResNet-50 [19] | 44 | 260 | 38.2 | 58.8 | 41.4 | 34.7 | 55.7 | 37.2 |
| ConvNeXt-T [32] | 48 | 262 | 44.2 | 66.6 | 48.3 | 40.1 | 63.3 | 42.8 |
| MLLA-T [15] | 44 | 255 | 46.8 | 69.5 | 51.5 | 42.1 | 66.4 | 45.0 |
| GrootVL-T [49] | 49 | 265 | 47.0 | 69.4 | 51.5 | 42.7 | 66.4 | 46.0 |
| VMamba-T [30] | 50 | 271 | 47.3 | 69.3 | 52.0 | 42.7 | 66.4 | 45.9 |
| Spatial-Mamba-T [47] | 46 | 216 | 47.6 | 69.6 | 52.3 | 42.9 | 66.5 | 46.2 |
| Swin-T [31] | 48 | 267 | 43.7 | 66.6 | 47.7 | 39.8 | 63.3 | 42.7 |
| CSWin-T [8] | 42 | 279 | 46.7 | 68.6 | 51.3 | 42.2 | 65.6 | 45.4 |
| BiFormer-S [57] | – | – | 47.8 | 69.8 | 52.3 | 43.2 | 66.8 | 46.5 |
| RMT-S [10] | 46 | 262 | 48.8 | **70.8** | 53.4 | 43.6 | **67.4** | **47.3** |
| PPMA-S | 46 | 263 | **49.2** | 70.7 | **54.0** | **43.8** | **67.4** | 47.1 |
| ResNet-101 [19] | 63 | 336 | 40.4 | 61.1 | 44.2 | 36.4 | 57.7 | 38.8 |
| ConvNeXt-S [32] | 70 | 348 | 45.4 | 67.9 | 50.0 | 41.8 | 65.2 | 45.1 |
| GrootVL-S [49] | 70 | 341 | 48.6 | 70.3 | 53.5 | 43.6 | 67.5 | 47.1 |
| VMamba-S [30] | 70 | 349 | 48.7 | 70.0 | 53.4 | 43.7 | 67.3 | 47.0 |
| Spatial-Mamba-S [47] | 63 | 315 | 49.2 | 70.8 | 54.2 | 44.0 | 67.9 | 47.5 |
| MLLA-S [15] | 63 | 319 | 49.2 | 71.5 | 53.9 | 44.2 | 68.5 | 47.2 |
| Swin-S [31] | 69 | 359 | 45.7 | 67.9 | 50.4 | 41.1 | 64.9 | 44.2 |
| CSWin-S [8] | 54 | 342 | 47.9 | 70.1 | 52.6 | 43.2 | 67.1 | 46.2 |
| BiFormer-B [57] | – | – | 48.6 | 70.5 | 53.8 | 43.7 | 67.6 | 47.1 |
| RMT-B [10] | 73 | 373 | 50.7 | 72.0 | 55.7 | 45.1 | 69.2 | 49.0 |
| PPMA-B | 73 | 374 | **51.1** | **72.5** | **55.9** | **45.5** | **69.7** | **49.1** |
| **Mask R-CNN 3× schedule** | | | | | | | | |
| ConvNeXt-S [32] | 70 | 348 | 47.9 | 70.0 | 52.7 | 42.9 | 66.9 | 46.2 |
| GrootVL-S [49] | 70 | 341 | 50.1 | 71.2 | 54.9 | 44.6 | 68.7 | 47.8 |
| VMamba-S [30] | 70 | 349 | 49.9 | 70.9 | 54.7 | 44.2 | 68.2 | 47.7 |
| MLLA-S [15] | 63 | 319 | 50.5 | 71.8 | 55.2 | 44.9 | 69.1 | 48.2 |
| Spatial-Mamba-S [47] | 63 | 315 | 50.6 | 71.5 | 55.4 | 44.7 | 68.6 | 48.2 |
| NAT-S [16] | 70 | 330 | 48.4 | 69.8 | 53.2 | 43.2 | 66.9 | 46.4 |
| Swin-S [31] | 69 | 359 | 48.5 | 70.2 | 53.5 | 43.3 | 67.3 | 46.6 |
| CSWin-S [8] | 54 | 342 | 50.0 | 71.3 | 54.7 | 44.5 | 68.4 | 47.7 |
| RMT-B [10] | 73 | 373 | 52.2 | 72.9 | 57.0 | 46.1 | **70.4** | 49.9 |
| PPMA-B | 73 | 374 | **52.6** | **73.3** | **57.5** | **46.3** | 70.3 | **50.2** |

Table 3: Semantic segmentation performance with UPerNet [48] segmentor on ADE20K val set. 'SS' and 'MS' represent single-scale and multi-scale testing, respectively.

| Backbone | #Param. (M) | FLOPs (G) | mIoU(%) SS | MS | Backbone | #Param. (M) | FLOPs (G) | mIoU(%) SS | MS |
|---|---|---|---|---|---|---|---|---|---|
| LocalVim-T [21] | 36 | 181 | 43.4 | 44.4 | BiFormer-S [57] | – | – | 49.8 | 50.8 |
| MSVMamba-M [36] | 42 | 875 | 45.1 | 45.4 | RMT-S [10] | 56 | 937 | 49.8 | 49.7 |
| NAT-M [16] | 50 | 900 | 45.1 | 46.4 | PPMA-S | 56 | 984 | **51.1** | **52.0** |
| RMT-T [10] | 43 | 977 | 48.0 | 48.8 | ResNet-101 [19] | 85 | 1030 | 42.9 | 44.0 |
| PPMA-T | 43 | 983 | **48.7** | **49.1** | ConvNeXt-S [32] | 82 | 1027 | 48.7 | 49.6 |
| ResNet-50 [19] | 67 | 953 | 42.1 | 42.8 | VMamba-S [30] | 82 | 1028 | 50.6 | 51.2 |
| ConvNeXt-T [32] | 60 | 939 | 46.0 | 46.7 | Spatial-Mamba-S [47] | 73 | 992 | 50.6 | 51.4 |
| VMamba-T [30] | 62 | 949 | 48.0 | 48.8 | GrootVL-S [49] | 82 | 1019 | 50.7 | 51.7 |
| 2DMamba-T [53] | 62 | 950 | 48.6 | 49.3 | Swin-S [31] | 81 | 1039 | 47.6 | 49.5 |
| GrootVL-T [49] | 60 | 941 | 48.5 | 49.4 | NAT-S [16] | 82 | 1010 | 48.0 | 49.5 |
| Spatial-Mamba-S [47] | 57 | 936 | 48.6 | 49.4 | MambaVision-S [17] | 84 | 1135 | 48.2 | – |
| Swin-T [31] | 60 | 945 | 44.4 | 45.8 | CSWin-S [8] | 65 | 1027 | 50.4 | 51.5 |
| MambaVision-T [17] | 55 | 945 | 46.6 | – | BiFormer-B [57] | – | – | 51.0 | 51.7 |
| NAT-T [16] | 58 | 934 | 47.1 | 48.4 | RMT-B [10] | 83 | 1051 | 52.0 | 52.1 |
| CSWin-S [8] | 60 | 959 | 49.3 | 50.7 | PPMA-B | 83 | 1137 | **52.3** | **53.0** |

## 5.2 Object Detection and Instance Segmentation on COCO

**Settings.** We evaluate our method for object detection and instance segmentation tasks on MSCOCO2017 [28] using the MMDetection library [2]. Following previous work [35], we initialize the backbone with ImageNet-1K pretrained weights and adopt Mask R-CNN [18] as the basic framework. The models are trained for 12 epochs (1× schedule) and 36 epochs with multi-scale inputs (3× schedule) using AdamW optimizer (batch size=16, learning rate=0.0001, weight decay=0.05).

**Results.** The results presented in Table 2 show that our model outperforms existing methods on most evaluation metrics. Under the same experimental settings, PPMA-T achieves a box mAP of **47.1%** and a mask mAP of **42.4%**, surpassing the SOTA Transformer-based backbone RMT-T [10] by **0.4%** and **0.3%** in the 1× schedule, respectively. Moreover, PPMA-B achieves a box mAP of **51.1%** and a mask mAP of **45.5%**, surpassing the SOTA SSM-based backbone MLLA-S [15] by **1.9%** and **1.3%** in the 1× schedule, respectively. Furthermore, PPMA-B maintains its superior performance under the 3× multi-scale training schedule.

## 5.3 Semantic Segmentation on ADE20K

**Settings.** We evaluate the semantic segmentation performance of our method on ADE20K [56] using the MMSegmentation library [4]. Following the settings in previous works [35], we initialize the backbone with ImageNet-1K pretrained weights and adopt UPerNet [48] as the basic framework. The input size of images is set to $512 \times 512$ and all models are trained for 160K iterations with AdamW optimizer (batch size=16, learning rate=$6 \times 10^{-5}$, weight decay=0.05).

**Results.** The semantic segmentation results are summarized in Table 3. Our method consistently outperforms previous methods under all settings. Compared to SOTA Transformer-based counterparts, PPMA-T/S/B surpass RMT-T/S/B by **0.7%/1.3%/0.3%** mIoU in the Single-Scale (SS) setting and **0.3%/2.3%/0.9%** mIoU in the Multi-Scale (MS) setting. Compared to SOTA SSM-based methods, PPMA-T/S/B surpass them by at least **3.6%/2.5%/1.6%** in SS mIoU, respectively.

## 5.4 Ablation Study

**Polyline Path Mask Design.** To verify the effectiveness of the proposed polyline path mask, we conduct an ablation study on ImageNet-1K and ADE20K using PPMA-T as the backbone. Under the same experimental settings, we compare various structured masks embedded into the softmax-based self-attention layers by Hadamard product, including: no mask (baseline), RMT decay mask [10], cross scan mask [30], Hilbert scan mask [27], V2H polyline path mask, and our final 2D polyline path mask. As shown in Fig. 6, our polyline path mask $L^{2D}$, compared to the RMT decay mask, can selectively capture the semantic continuity in the image. Compared to the cross scan mask and Hilbert scan mask, the polyline path mask better preserves the spatial relationships between 2D tokens, alleviating the long-range forgetting issue. Experimental results in Table 4 show that the 2D

Table 4: Ablation study of structured mask designs in PPMA-T on ImageNet-1K and ADE20K.

| Structured Mask | #Param. (M) | FLOPs (G) | Throughput (imgs/s) | Top-1 (%) | mIoU SS (%) |
|---|---|---|---|---|---|
| Baseline (w/o mask) | 14.33 | 2.65 | 1779 | 82.28 | 47.78 |
| + RMT Decay Mask | 14.33 | 2.65 | 1650 | 82.35 | 48.01 |
| + Cross Scan Mask | 14.34 | 2.71 | 1100 | 82.44 | 48.14 |
| + Hilbert Scan Mask | 14.34 | 2.71 | 1091 | 82.44 | 48.14 |
| + V2H Polyline Path Mask | 14.34 | 2.71 | 1203 | 82.44 | 48.57 |
| + 2D Polyline Path Mask | 14.34 | 2.71 | 1034 | **82.60** | **48.73** |
| Shared Decay factors ($\alpha_{i,j} = \beta_{i,j}$) | 14.33 | 2.71 | 1124 | 82.37 | 48.27 |
| Different Decay factors ($\alpha_{i,j} \neq \beta_{i,j}$) | 14.34 | 2.71 | 1034 | **82.60** | **48.73** |

(a) Input Image   (b) W/o Mask   (c) RMT Decay Mask   (d) Cross Scan Mask   (e) Hilbert Scan Mask   (f) Polyline Path Mask

Figure 6: Illustration of various structured masks.

(a) Input Image   (b) Horizontal Decay Factor $\boldsymbol{\alpha}$   (c) Vertical Decay Factor $\boldsymbol{\beta}$   (d) Original Attention Map   (e) Polyline Path Mask $\boldsymbol{L}^{2D}$   (f) Polyline Path Masked Attention Map

Figure 7: Visualizations of the decay factors and the polyline path masked attention maps of the well-trained PPMA model. In each input image, the query token is marked by a red box.

polyline path mask $\boldsymbol{L}^{2D}$ boosts the baseline by **0.32%** top-1 accuracy on ImageNet-1K and **0.95%** SS mIoU on ADE20K, respectively. Visualization results in Fig. 7 further demonstrate that our 2D polyline path mask $\boldsymbol{L}^{2D}$ effectively suppresses the falsely highlighted areas in the original attention maps. More visualizations and detailed discussions are provided in Fig. 12 and Sec. C.1.

**Horizontal and Vertical Decay Factors.** In our model, we employ different decay factors ($\alpha_{i,j} \neq \beta_{i,j}$) to capture semantic similarity between adjacent tokens along horizontal and vertical directions, respectively. As illustrated in Fig. 7 (b) and (c), the learned decay factors $\alpha$ and $\beta$ effectively capture semantic continuity in horizontal and vertical directions, respectively. Table 4 shows that replacing different decay factors with a shared decay factor ($\alpha_{i,j} = \beta_{i,j}$) results in a significant performance drop, highlighting the importance of modeling horizontal and vertical decay factors separately.

## 6 Conclusion

In this paper, we argue that the key component of Mamba2 model is its structured mask, which explicitly encodes the spatial distance information through the recursive propagation mechanism and captures the semantic continuity in sequences through the selective mechanism. Building on this insight, we propose to extend the structured mask from 1D text sequences to 2D images. To this end, we propose a novel 2D polyline path scanning strategy with its corresponding structured mask tailed for images. To achieve SOTA performance on high-level vision tasks, we integrate the polyline path mask into the powerful self-attention mechanism of ViTs.

**Limitations.** Although the proposed efficient algorithm optimizes the integration complexity, it inevitably incurs additional GPU memory occupation and lower throughput, as shown in Table 4. We plan to alleviate this limitation through further engineering optimizations, such as CUDA-based or Triton-based implementations, in the future work.

## Acknowledgments

We would like to thank all anonymous reviewers for their constructive suggestions for improving this paper. This work was supported in part by the National Key R&D Program of China under Grant 2024YFA1012000; in part by the Major Key Project of PCL under Grant PCL2024A06; in part by Tianyuan Fund for Mathematics of the National Natural Science Foundation of China (Grant No. 12426105); in part by the China NSFC projects under contract 62476214.

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

# Supplementary Material

## A  Efficient Computation Theory for Polyline Path Mask Applications

### A.1  Notations

Following Mamba2 [6], we employ a large number of notations both for clarity and as a central tool in stating and proving our theorems, including:

- **Dimensions.** We generally use $N$, $H$, $W$, $C$, $D$ as the superscript letters of $\mathbb{R}$ to denote the sequence length, the height of the feature map, the width of the feature map, channel number, and hidden state dimension, respectively. The sequence length of the 2D feature map (i.e., the number of tokens) is $N = H \times W$.

- **Matrices and Tensors.** Following convention, we use non-bolded lowercase letters, bolded lowercase letters, bolded uppercase letters, and bolded calligraphy letters to denote scalars, vectors, matrices, and 3D or higher dimensional tensors, respectively.

- **Indexing.** We use indexing $i : j$ to refer to the range $i+1, i+2, \ldots, j$ when $i < j$ and $i, i-1, \ldots, j+1$ when $i > j$. For example, for any scalar $a$, we let $a_{i:j}$ for $i < j$ denote the sequence $(a_{i+1}, a_{i+2}, \ldots, a_j)$. For shorthand, we let $a_{i:j}^{\times}$ for $i < j$ denote the product $a_{i+1} \times a_{i+2} \times \cdots \times a_j{}^4$. We let $a_{j:i}^{\times} = a_{i:j}^{\times}$ for $j > i$.

- **Tensor Unfolding.** We use the operator $\mathtt{vec}(\cdot)$ to vectorize a matrix by stacking its columns and the operator $\mathtt{unvec}(\cdot)$ as its inverse operation. We use the operator $\mathtt{unfold}(\cdot)$ to unfolds a 4D tensor $\boldsymbol{\mathcal{L}} \in \mathbb{R}^{H \times W \times H \times W}$ to a 2D matrix $\boldsymbol{L} \in \mathbb{R}^{HW \times HW}$, where $[\boldsymbol{L}]_{(i-1)\times W+j,(k-1)\times W+l} = \boldsymbol{\mathcal{L}}_{i,j,k,l}$, and the operator $\mathtt{fold}(\cdot)$ as its inverse operation.

- **Tensor Multiplication.** For 2D matrices, we use the symbol $\times$ to denote the matrix multiplication and the symbol $\odot$ to denote the Hadamard (element-wise) multiplication. For multiplication operations involving 3D or higher dimensional tensors, we use the Einstein summation notation $\mathtt{einsum}(\cdot)$ to denote the tensor multiplication on the given dimension, which is commonly used in modern tensor libraries such as PyTorch. For example, $\mathtt{einsum}('\mathtt{nc}, \mathtt{mc} \to \mathtt{nm}', \boldsymbol{Q}, \boldsymbol{K})$ denotes the matrix multiplication $\boldsymbol{Q} \times \boldsymbol{K}^{\top}$.

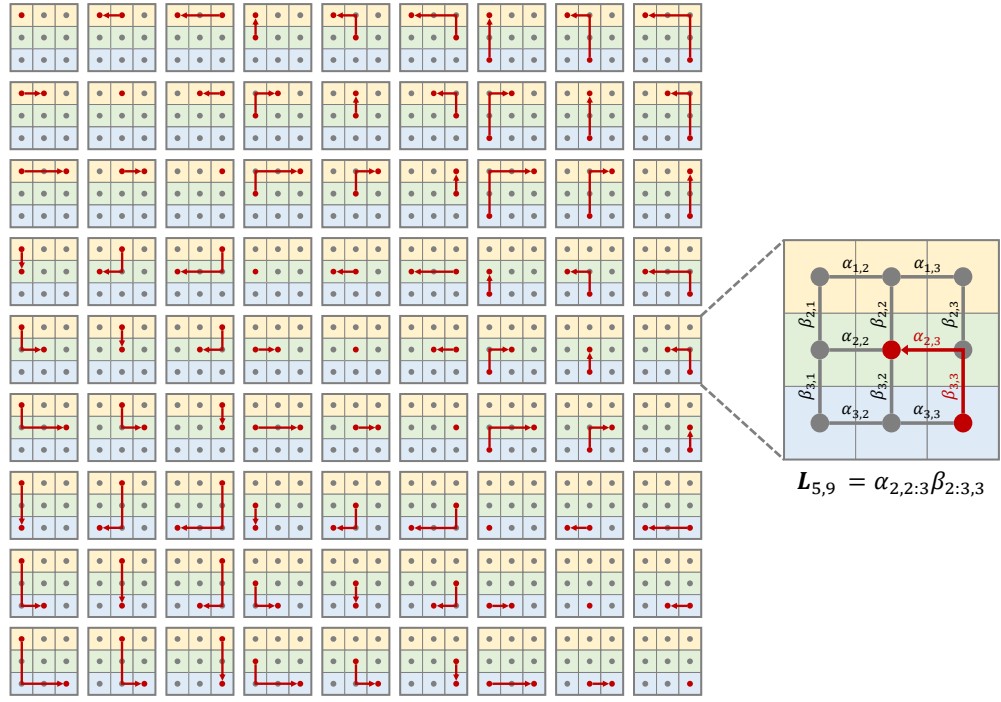

Figure 8: An illustration of the V2H polyline path scanning on a $3 \times 3$ grid (with a total of 9 tokens). There are 81 scanning paths. Each scanning path (red polyline) corresponds to a decay weight in the polyline path mask $\boldsymbol{L}$.

## A.2 Definition of Polyline Path Mask

For each token pair $(\boldsymbol{x}_{i,j}, \boldsymbol{x}_{k,l})$ in the 2D grid, the decay weight of the vertical-then-horizontal (V2H) polyline path from $\boldsymbol{x}_{i,j}$ to $\boldsymbol{x}_{k,l}$ is defined as $\boldsymbol{\mathcal{L}}_{i,j,k,l}$, which is the product of all decay factors along that path, i.e.,

$$\boldsymbol{\mathcal{L}}_{i,j,k,l} = \alpha_{i,j:l}\beta_{i:k,l}, \text{ where } \alpha_{i,j:l} = \begin{cases} \prod_{n=j+1}^{l} \alpha_{i,n} & j < l \\ 1 & j = l \\ \prod_{n=l+1}^{j} \alpha_{i,n} & j > l \end{cases}, \ \beta_{i:k,l} = \begin{cases} \prod_{n=i+1}^{k} \beta_{n,l} & i < k \\ 1 & i = k \\ \prod_{n=k+1}^{i} \beta_{n,l} & i > k \end{cases}, \quad (13)$$

where $\alpha_{i,j:l}$ and $\beta_{i:k,l}$ are horizontal and vertical decay factors bounded in the range $[0,1]$. For convenience, we unfold the 4D tensor $\boldsymbol{\mathcal{L}} \in \mathbb{R}^{H \times W \times H \times W}$ into a 2D matrix as the polyline path mask $\boldsymbol{L} \in \mathbb{R}^{HW \times HW}$, i.e.,

$$\boldsymbol{L} = \text{unfold}(\boldsymbol{\mathcal{L}}). \quad (14)$$

$$\boldsymbol{L} = \begin{bmatrix} \begin{matrix} \alpha_{1,1:1}\beta_{1:1,1} & \alpha_{1,1:2}\beta_{1:1,2} & \cdots & \alpha_{1,1:W}\beta_{1:1,W} \\ \alpha_{1,2:1}\beta_{1:1,1} & \alpha_{1,2:2}\beta_{1:1,2} & \cdots & \alpha_{1,2:W}\beta_{1:1,W} \\ \vdots & \vdots & \ddots & \vdots \\ \alpha_{1,W:1}\beta_{1:1,1} & \alpha_{1,W:2}\beta_{1:1,2} & \cdots & \alpha_{1,W:W}\beta_{1:1,W} \end{matrix} & \cdots & \begin{matrix} \alpha_{1,1:1}\beta_{1:H,1} & \alpha_{1,1:2}\beta_{1:H,2} & \cdots & \alpha_{1,1:W}\beta_{1:H,W} \\ \alpha_{1,2:1}\beta_{1:H,1} & \alpha_{1,2:2}\beta_{1:H,2} & \cdots & \alpha_{1,2:W}\beta_{1:H,W} \\ \vdots & \vdots & \ddots & \vdots \\ \alpha_{1,W:1}\beta_{1:H,1} & \alpha_{1,W:2}\beta_{1:H,2} & \cdots & \alpha_{1,W:W}\beta_{1:H,W} \end{matrix} \\ \vdots & \ddots & \vdots \\ \begin{matrix} \alpha_{H,1:1}\beta_{H:1,1} & \alpha_{H,2:1}\beta_{H:1,1} & \cdots & \alpha_{H,W:1}\beta_{H:1,1} \\ \alpha_{H,1:2}\beta_{H:1,2} & \alpha_{H,2:2}\beta_{H:1,2} & \cdots & \alpha_{H,W:2}\beta_{H:1,2} \\ \vdots & \vdots & \ddots & \vdots \\ \alpha_{H,1:W}\beta_{H:1,W} & \alpha_{H,2:W}\beta_{H:1,W} & \cdots & \alpha_{H,W:W}\beta_{H:1,W} \end{matrix} & \cdots & \begin{matrix} \alpha_{H,1:1}\beta_{H:H,1} & \alpha_{H,1:2}\beta_{H:H,2} & \cdots & \alpha_{H,1:W}\beta_{H:H,W} \\ \alpha_{H,2:1}\beta_{H:H,1} & \alpha_{H,2:2}\beta_{H:H,2} & \cdots & \alpha_{H,2:W}\beta_{H:H,W} \\ \vdots & \vdots & \ddots & \vdots \\ \alpha_{H,W:1}\beta_{H:H,1} & \alpha_{H,W:2}\beta_{H:H,2} & \cdots & \alpha_{H,W:W}\beta_{H:H,W} \end{matrix} \end{bmatrix}$$
$$(15)$$

An intuitive example illustrating the polyline path scanning on a $3 \times 3$ grid is presented in Fig. 8. For the 9 tokens in the 2D grid, there are 81 V2H scanning paths connecting them. The V2H scanning path between each token pair is marked by the red polyline, which corresponds to a decay weight in

---

[4] In some contexts, it is always clear that the notation $a_{i:j}$ means $a_{i:j}^{\times}$, and the superscript is omitted.

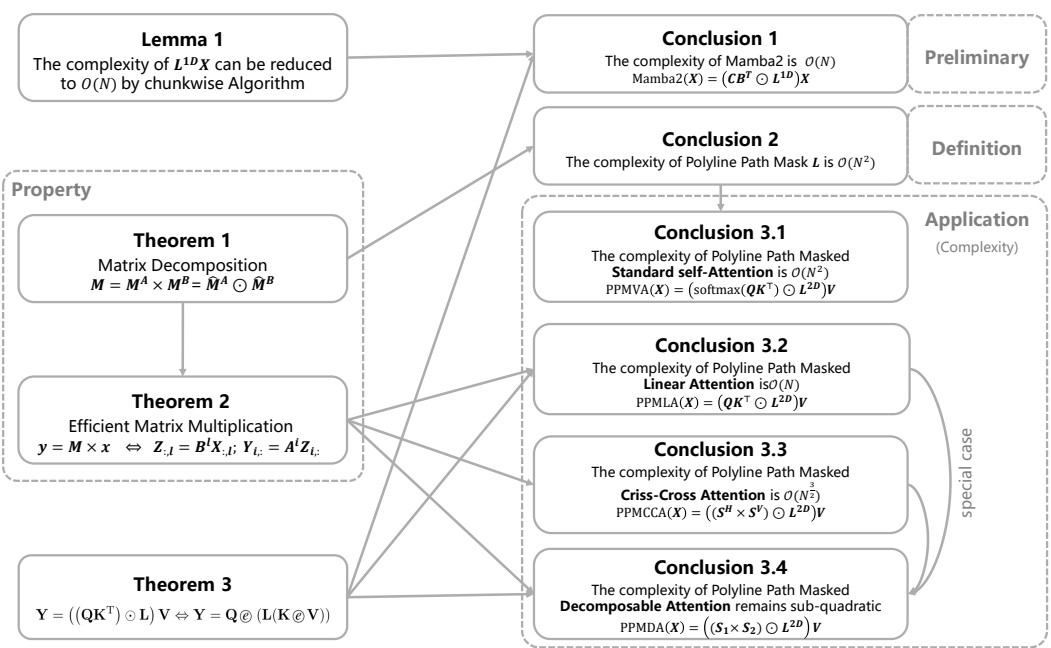

Figure 9: An overall illustration of the efficient computation theory and corresponding applications.

the polyline path mask $L$. The V2H polyline path mask $L \in \mathbb{R}^{9 \times 9}$, constructed based on the scanning paths in Fig. 8, is defined as:

$$
L = \begin{bmatrix}
\alpha_{1,1:1}\beta_{1:1,1} & \alpha_{1,1:2}\beta_{1:1,2} & \alpha_{1,1:3}\beta_{1:1,3} & \alpha_{1,1:1}\beta_{1:2,1} & \alpha_{1,1:2}\beta_{1:2,2} & \alpha_{1,1:3}\beta_{1:2,3} & \alpha_{1,1:1}\beta_{1:3,1} & \alpha_{1,1:2}\beta_{1:3,2} & \alpha_{1,1:3}\beta_{1:3,3} \\
\alpha_{1,2:1}\beta_{1:1,1} & \alpha_{1,2:2}\beta_{1:1,2} & \alpha_{1,2:3}\beta_{1:1,3} & \alpha_{1,2:1}\beta_{1:2,1} & \alpha_{1,2:2}\beta_{1:2,2} & \alpha_{1,2:3}\beta_{1:2,3} & \alpha_{1,2:1}\beta_{1:3,1} & \alpha_{1,2:2}\beta_{1:3,2} & \alpha_{1,2:3}\beta_{1:3,3} \\
\alpha_{1,3:1}\beta_{1:1,1} & \alpha_{1,3:2}\beta_{1:1,2} & \alpha_{1,3:3}\beta_{1:1,3} & \alpha_{1,3:1}\beta_{1:2,1} & \alpha_{1,3:2}\beta_{1:2,2} & \alpha_{1,3:3}\beta_{1:2,3} & \alpha_{1,3:1}\beta_{1:3,1} & \alpha_{1,3:2}\beta_{1:3,2} & \alpha_{1,3:3}\beta_{1:3,3} \\
\alpha_{2,1:1}\beta_{2:1,1} & \alpha_{2,1:2}\beta_{2:1,2} & \alpha_{2,1:3}\beta_{2:1,3} & \alpha_{2,1:1}\beta_{2:2,1} & \alpha_{2,1:2}\beta_{2:2,2} & \alpha_{2,1:3}\beta_{2:2,3} & \alpha_{2,1:1}\beta_{2:3,1} & \alpha_{2,1:2}\beta_{2:3,2} & \alpha_{2,1:3}\beta_{2:3,3} \\
\alpha_{2,2:1}\beta_{2:1,1} & \alpha_{2,2:2}\beta_{2:1,2} & \alpha_{2,2:3}\beta_{2:1,3} & \alpha_{2,2:1}\beta_{2:2,1} & \alpha_{2,2:2}\beta_{2:2,2} & \alpha_{2,2:3}\beta_{2:2,3} & \alpha_{2,2:1}\beta_{2:3,1} & \alpha_{2,2:2}\beta_{2:3,2} & \alpha_{2,2:3}\beta_{2:3,3} \\
\alpha_{2,3:1}\beta_{2:1,1} & \alpha_{2,3:2}\beta_{2:1,2} & \alpha_{2,3:3}\beta_{2:1,3} & \alpha_{2,3:1}\beta_{2:2,1} & \alpha_{2,3:2}\beta_{2:2,2} & \alpha_{2,3:3}\beta_{2:2,3} & \alpha_{2,3:1}\beta_{2:3,1} & \alpha_{2,3:2}\beta_{2:3,2} & \alpha_{2,3:3}\beta_{2:3,3} \\
\alpha_{3,1:1}\beta_{3:1,1} & \alpha_{3,1:2}\beta_{3:1,2} & \alpha_{3,1:3}\beta_{3:1,3} & \alpha_{3,1:1}\beta_{3:2,1} & \alpha_{3,1:2}\beta_{3:2,2} & \alpha_{3,1:3}\beta_{3:2,3} & \alpha_{3,1:1}\beta_{3:3,1} & \alpha_{3,1:2}\beta_{3:3,2} & \alpha_{3,1:3}\beta_{3:3,3} \\
\alpha_{3,2:1}\beta_{3:1,1} & \alpha_{3,2:2}\beta_{3:1,2} & \alpha_{3,2:3}\beta_{3:1,3} & \alpha_{3,2:1}\beta_{3:2,1} & \alpha_{3,2:2}\beta_{3:2,2} & \alpha_{3,2:3}\beta_{3:2,3} & \alpha_{3,2:1}\beta_{3:3,1} & \alpha_{3,2:2}\beta_{3:3,2} & \alpha_{3,2:3}\beta_{3:3,3} \\
\alpha_{3,3:1}\beta_{3:1,1} & \alpha_{3,3:2}\beta_{3:1,2} & \alpha_{3,3:3}\beta_{3:1,3} & \alpha_{3,3:1}\beta_{3:2,1} & \alpha_{3,3:2}\beta_{3:2,2} & \alpha_{3,3:3}\beta_{3:2,3} & \alpha_{3,3:1}\beta_{3:3,1} & \alpha_{3,3:2}\beta_{3:3,2} & \alpha_{3,3:3}\beta_{3:3,3}
\end{bmatrix}
\tag{16}
$$

## A.3 Theorems and Proofs

In this section, we present three theorems and their proofs, which will be used to optimize the computational complexity in the following applications A.7. Specifically, we present a decomposition theorem 1 for matrices structured as the polyline path mask $L$. Based on Theorem 1, we present an efficient matrix multiplication theorem 2 for performing multiplication on the polyline path mask $L$. Then, we present an equivalent computation theorem 3 for the masked linear attention. An overview illustration is provided in Fig. 9, which summarizes the theorems and their corresponding applications in polyline path masked attention.

Note that the polyline path mask $L$ defined in Eq. (15) is a matrix with special structures. Here, we present a decomposition theorem for matrices structured as $L$.

**Theorem 1** (Matrix Decomposition)**.** *For any matrix* $M \in \mathbb{R}^{HW \times HW}$ *and* $\mathcal{M} = \mathrm{fold}(M)$, *if for* $\forall i, j, k, l$, $\exists A^i \in \mathbb{R}^{W \times W}$ *and* $B^l \in \mathbb{R}^{H \times H}$, *s.t.*, $\mathcal{M}_{i,j,k,l} = \left[A^i\right]_{j,l} \times \left[B^l\right]_{i,k}$, *then* $M$ *can be decomposed as:*

$$
M = M^A \times M^B = \hat{M}^A \odot \hat{M}^B,
\tag{17}
$$

*where* $M^A, M^B, \hat{M}^A, \hat{M}^B \in \mathbb{R}^{HW \times HW}$, *which satisfy*

$$
M^A = \mathrm{unfold}(\mathcal{M}^A), \ M^B = \mathrm{unfold}(\mathcal{M}^B), \ s.t., \ \mathcal{M}^A_{i,:,k,:} = \begin{cases} A^i & k=i \\ 0 & k \neq i \end{cases}, \ \mathcal{M}^B_{:,j,:,l} = \begin{cases} B^l & j=l \\ 0 & j \neq l \end{cases},
\tag{18}
$$

$$
\hat{M}^A = \mathrm{unfold}(\hat{\mathcal{M}}^A), \ \hat{M}^B = \mathrm{unfold}(\hat{\mathcal{M}}^B), \ s.t., \ \hat{\mathcal{M}}^A_{i,:,k,:} = A^i, \ \hat{\mathcal{M}}^B_{:,j,:,l} = B^l.
\tag{19}
$$

*Proof.* Let us first prove $\boldsymbol{M}^A \times \boldsymbol{M}^B = \boldsymbol{M}$ in Eq. (17). For clarity, let $i = \lfloor u/W \rfloor + 1$, $j = u \bmod W$, $k = \lfloor v/W \rfloor + 1$, $l = v \bmod W$, $m = \lfloor w/W \rfloor + 1$, $n = w \bmod W$. For $u = 1 : HW$ and $v = 1 : HW$, we have

$$
\begin{aligned}
\sum_{w=1}^{HW} \boldsymbol{M}_{u,w}^A \boldsymbol{M}_{w,v}^B &= \sum_{m=1}^{H} \sum_{n=1}^{W} \boldsymbol{\mathcal{M}}_{i,j,m,n}^A \boldsymbol{\mathcal{M}}_{m,n,k,l}^B \\
&= \sum_{m=1}^{H} \left( \boldsymbol{\mathcal{M}}_{i,j,m,l}^A \boldsymbol{\mathcal{M}}_{m,l,k,l}^B + \sum_{n=1,n\neq l}^{W} \boldsymbol{\mathcal{M}}_{i,j,m,n}^A \boldsymbol{\mathcal{M}}_{m,n,k,l}^B \right) \\
&= \sum_{m=1}^{H} \boldsymbol{\mathcal{M}}_{i,j,m,l}^A \boldsymbol{\mathcal{M}}_{m,l,k,l}^B \\
&= \boldsymbol{\mathcal{M}}_{i,j,i,l}^A \boldsymbol{\mathcal{M}}_{i,l,k,l}^B + \sum_{m=1,m\neq i}^{H} \boldsymbol{\mathcal{M}}_{i,j,m,l}^A \boldsymbol{\mathcal{M}}_{m,l,k,l}^B \\
&= \boldsymbol{\mathcal{M}}_{i,j,i,l}^A \boldsymbol{\mathcal{M}}_{i,l,k,l}^B = \boldsymbol{A}_{j,l}^i \boldsymbol{B}_{i,k}^l = \boldsymbol{\mathcal{M}}_{i,j,k,l} \\
&= \boldsymbol{M}_{u,v}.
\end{aligned}
\tag{20}
$$

According to Eq. (19), we have

$$
\hat{\boldsymbol{M}}_{u,v}^A \hat{\boldsymbol{M}}_{u,v}^B = \hat{\boldsymbol{\mathcal{M}}}_{i,j,k,l}^A \hat{\boldsymbol{\mathcal{M}}}_{i,j,k,l}^B = \left[\boldsymbol{A}^i\right]_{j,l} \left[\boldsymbol{B}^l\right]_{i,k} = \boldsymbol{\mathcal{M}}_{i,j,k,l} = \boldsymbol{M}_{u,v}.
\tag{21}
$$

Thus, Theorem 1 is proven. $\qquad\square$

For matrices of the form given in Eq. (18), when performing multiplication operations, we have:

**Theorem 2** (Efficient Matrix Multiplication). *For matrices $\boldsymbol{M}^A$, $\boldsymbol{M}^B$ defined in Eq. (18), $\forall \boldsymbol{x} \in \mathbb{R}^{HW}$, the following equation holds:*

$$
\boldsymbol{y} = \boldsymbol{M}^A \times \boldsymbol{M}^B \times \boldsymbol{x} \quad \Leftrightarrow \quad \boldsymbol{Z}_{:,l} = \boldsymbol{B}^l \times \boldsymbol{X}_{:,l}, \; \boldsymbol{Y}_{i,:} = \boldsymbol{A}^i \times \boldsymbol{Z}_{i,:},
\tag{22}
$$

*where $\boldsymbol{y} \in \mathbb{R}^{HW}$, $\boldsymbol{X} = \mathrm{unvec}(\boldsymbol{x}) \in \mathbb{R}^{H\times W}$, $\boldsymbol{Y} = \mathrm{unvec}(\boldsymbol{y}) \in \mathbb{R}^{H\times W}$, $\boldsymbol{Z} \in \mathbb{R}^{H\times W}$.*

*Proof.* The left part of Eq. (22) can be calculated by $\boldsymbol{z} = \boldsymbol{M}^B \times \boldsymbol{x}$ and $\boldsymbol{y} = \boldsymbol{M}^A \times \boldsymbol{z}$. For clarity, let $i = \lfloor u/W \rfloor + 1$, $j = u \bmod W$, $k = \lfloor v/W \rfloor + 1$, $l = v \bmod W$. For $u = 1 : HW$, we have

$$
\begin{aligned}
\boldsymbol{z}_u &= \sum_{v=1}^{HW} \boldsymbol{M}_{u,v}^B \times \boldsymbol{x}_v = \sum_{k=1}^{H} \sum_{l=1}^{W} \boldsymbol{\mathcal{M}}_{i,j,k,l}^B \boldsymbol{X}_{k,l} \\
&= \sum_{k=1}^{H} \boldsymbol{\mathcal{M}}_{i,j,k,j}^B \boldsymbol{X}_{k,j} + \sum_{k=1}^{H} \sum_{l=1,l\neq j}^{W} \boldsymbol{\mathcal{M}}_{i,j,k,l}^B \boldsymbol{X}_{k,l} \\
&= \sum_{k=1}^{H} \boldsymbol{\mathcal{M}}_{i,j,k,j}^B \boldsymbol{X}_{k,j} = \sum_{k=1}^{H} \boldsymbol{B}_{i,k}^j \boldsymbol{X}_{k,j} \\
&= \boldsymbol{Z}_{i,j}.
\end{aligned}
\tag{23}
$$

The final results in Eq. (23) is equivalent to $\boldsymbol{Z}_{:,j} = \boldsymbol{B}^j \times \boldsymbol{X}_{:,j}$. Then, for $u = 1 : HW$, we have

$$
\begin{aligned}
\boldsymbol{y}_u &= \sum_{v=1}^{HW} \boldsymbol{M}_{u,v}^A \times \boldsymbol{z}_v = \sum_{k=1}^{H} \sum_{l=1}^{W} \boldsymbol{\mathcal{M}}_{i,j,k,l}^A \boldsymbol{Z}_{k,l} \\
&= \sum_{l=1}^{W} \boldsymbol{\mathcal{M}}_{i,j,i,l}^A \boldsymbol{Z}_{i,l} + \sum_{k=1,k\neq i}^{H} \sum_{l=1}^{W} \boldsymbol{\mathcal{M}}_{i,j,k,l}^A \boldsymbol{Z}_{k,l} \\
&= \sum_{l=1}^{W} \boldsymbol{\mathcal{M}}_{i,j,i,l}^A \boldsymbol{Z}_{i,l} = \sum_{l=1}^{W} \boldsymbol{A}_{j,l}^i \boldsymbol{Z}_{i,l} \\
&= \boldsymbol{Y}_{i,j}.
\end{aligned}
\tag{24}
$$

The final results in Eq. (24) is equivalent to $\boldsymbol{Y}_{i,:} = \boldsymbol{A}^i \times \boldsymbol{Z}_{i,:}$. Thus, Theorem 2 is proven. $\qquad\square$

**Theorem 3.** *For any matrices $\boldsymbol{Q}, \boldsymbol{K} \in \mathbb{R}^{N \times D}, \boldsymbol{L} \in \mathbb{R}^{N \times N}, \boldsymbol{V} \in \mathbb{R}^{N \times C}$, the following equation holds:*

$$\boldsymbol{Y} = \left((\boldsymbol{Q}\boldsymbol{K}^\top) \odot \boldsymbol{L}\right)\boldsymbol{V} \quad \Leftrightarrow \quad \mathcal{K}^V = \texttt{einsum}\left('\texttt{nd}, \texttt{nc} \to \texttt{ndc}', \boldsymbol{K}, \boldsymbol{V}\right)$$

$$\mathcal{L}^{KV} = \texttt{einsum}\left('\texttt{mn}, \texttt{ndc} \to \texttt{mdc}', \boldsymbol{L}, \mathcal{K}^V\right) \quad (25)$$

$$\boldsymbol{Y} = \texttt{einsum}\left('\texttt{md}, \texttt{mdc} \to \texttt{mc}', \boldsymbol{Q}, \mathcal{L}^{KV}\right),$$

*Proof.* 1) the left part of the Eq. (25) can be rewritten as:

$$\boldsymbol{S} = \boldsymbol{Q}\boldsymbol{K}^\top, \quad \text{where} \quad \boldsymbol{S}_{m,n} = \sum_{d=1}^{D} \boldsymbol{Q}_{m,d}\boldsymbol{K}_{n,d},$$

$$\boldsymbol{Y}_{m,c} = \sum_{n=1}^{N} \boldsymbol{L}_{m,n}\boldsymbol{S}_{m,n}\boldsymbol{V}_{n,c} = \sum_{n=1}^{N}\left(\boldsymbol{L}_{m,n}\sum_{d=1}^{D}\boldsymbol{Q}_{m,d}\boldsymbol{K}_{n,d}\right)\boldsymbol{V}_{n,c}. \quad (26)$$

2) the right part of the Eq. (25) can be rewritten as:

$$\mathcal{K}^V_{n,d,c} = \boldsymbol{K}_{n,d}\boldsymbol{V}_{n,c},$$

$$\mathcal{L}^{KV}_{m,d,c} = \sum_{n=1}^{N}\boldsymbol{L}_{m,n}\mathcal{K}^V_{n,d,c} = \sum_{n=1}^{N}\boldsymbol{L}_{m,n}\boldsymbol{K}_{n,d}\boldsymbol{V}_{n,c},$$

$$\boldsymbol{Y}_{m,c} = \sum_{d=1}^{D}\boldsymbol{Q}_{m,d}\mathcal{L}^{KV}_{m,d,c} = \sum_{d=1}^{D}\left(\boldsymbol{Q}_{m,d}\sum_{n=1}^{N}\boldsymbol{L}_{m,n}\boldsymbol{K}_{n,d}\boldsymbol{V}_{n,c}\right) \quad (27)$$

$$= \sum_{d=1}^{D}\sum_{n=1}^{N}\boldsymbol{Q}_{m,d}\boldsymbol{L}_{m,n}\boldsymbol{K}_{n,d}\boldsymbol{V}_{n,c}$$

$$= \sum_{n=1}^{N}\left(\boldsymbol{L}_{m,n}\sum_{d=1}^{D}\boldsymbol{Q}_{m,d}\boldsymbol{K}_{n,d}\right)\boldsymbol{V}_{n,c}.$$

Thus, Theorem 3 is proven. Here, the computational complexity of the left part of Eq. (25) is $\mathcal{O}(N^2)$. The computational complexity of the first and third lines in right part of Eq. (25) is $\mathcal{O}(N)$. And the computational complexity of the second line in right part of Eq. (25) is $\mathcal{O}(N^2)$. Thus, if we can reduce the complexity of computing $\mathcal{L}^{KV}$ from $\mathcal{O}(N^2)$ to $\mathcal{O}(N)$, then the complexity of computing $\boldsymbol{Y} = \left((\boldsymbol{Q}\boldsymbol{K}^\top) \odot \boldsymbol{L}\right)\boldsymbol{V}$ can be reduced to $\mathcal{O}(N)$. $\qquad\square$

### A.4 Preliminaries: Complexity Analysis of Mamba2 Attention Form

In this section, we present the efficient algorithm proposed in Mamba2 [6] for its attention form, achieving a computational complexity of $\mathcal{O}(N)$.

**Mamba2's Attention Form.** Mamba2's attention form (i.e., structured masked attention) given by the SSD framework [6] is formulated as:

$$\boldsymbol{Y} = \left(\boldsymbol{C}\boldsymbol{B}^\top \odot \boldsymbol{L}^{1D}\right)\boldsymbol{X}, \qquad \boldsymbol{L}^{1D}_{ij} = a_{i:j} = \begin{cases} a_i \times \cdots \times a_{j+1} & i > j \\ 1 & i = j \\ 0 & i < j \end{cases}, \quad (28)$$

where $\boldsymbol{X}, \boldsymbol{Y} \in \mathbb{R}^{N \times C}$ are the input and output sequences, respectively, $\boldsymbol{B}, \boldsymbol{C} \in \mathbb{R}^{N \times D}$ are input-dependent parameters learned by multilayer perceptron (MLP) layers. The 1D structured mask $\boldsymbol{L}^{1D} \in \mathbb{R}^{N \times N}$ is a 1-semiseparable matrix [6], and the scalar $a_i$ serves as a decay factor bounded in the range $[0, 1]$. In Mamba2, parameters $\boldsymbol{C}$ and $\boldsymbol{B}$ in Eq. (28) correspond to the query $\boldsymbol{Q}$ and key $\boldsymbol{K}$ in ViTs, respectively. Therefore, Eq. (28) reveals that the selective state transition function in Mamba2 is equivalent to the Hadamard product of a linear attention map $\boldsymbol{C}\boldsymbol{B}^\top$ and a 1D structured mask $\boldsymbol{L}^{1D}$.

**Naive Computation.** As defined in Eq. (28), the straightforward computation of structured masked attention has a complexity of $\mathcal{O}(N^2)$. In contrast, the complexity of linear attention $\boldsymbol{Y} = (\boldsymbol{C}\boldsymbol{B}^\top)\boldsymbol{X} = \boldsymbol{C}(\boldsymbol{B}^\top\boldsymbol{X})$ can be reduced from $\mathcal{O}(N^2)$ to $\mathcal{O}(N)$ by the associative property of matrix multiplication. However, this approach is not directly applicable to Eq. (28) because of the introduction of the Hadamard product.

**Lemma 1.** *Let $L \in \mathbb{R}^{N \times N}$ be a 1-semiseparable matrix and $X \in \mathbb{R}^{N \times C}$ be a matrix, the complexity of computing $Y = LX$ can be reduced from $\mathcal{O}(N^2)$ to $\mathcal{O}(N)$ by using the chunkwise algorithm in Mamba2 [6].*

**Efficient Computation.** Based on Theorem 3, Eq. (28) can be computed as follows:

$$
\begin{aligned}
\mathcal{B}^X &= \texttt{einsum}\left('\texttt{nd},\texttt{nc} \to \texttt{ndc}', B, X\right) \\
\mathcal{L}^{BX} &= \texttt{einsum}\left('\texttt{mn},\texttt{ndc} \to \texttt{mdc}', L^{1D}, \mathcal{B}^X\right) \\
Y &= \texttt{einsum}\left('\texttt{md},\texttt{mdc} \to \texttt{mc}', C, \mathcal{L}^{BX}\right).
\end{aligned}
\tag{29}
$$

Note that $L^{1D}$ is a 1-semiseparable matrix, and the second line of Eq. (29) can be reformulated as a matrix multiplication. Therefore, by applying Lemma 1, the complexity of computing $\mathcal{L}^{BX}$ can be reduced from $\mathcal{O}(N^2)$ to $\mathcal{O}(N)$. Moreover, the complexity of computing $L^{1D}$ is $\mathcal{O}(N)$. Consequently, the overall computational complexity of Eq. (28) is $\mathcal{O}(N)$.

### A.5 Complexity Analysis of Polyline Path Mask

In this section, we analyze the complexity of computing the polyline path mask $L$, as stated in Corollary 1, with detailed explanation.

**Corollary 1** (Mask Complexity). *The complexity of directly computing polyline path mask $L$ via Eq.(16) and (14) is $\mathcal{O}(N^{\frac{5}{2}})$, which can be reduced to $\mathcal{O}(N^2)$ by applying Theorem 1, where $N = H \times W$.*

**Naive Computation.** According to the definition in Eq. (15), the polyline path mask $L \in \mathbb{R}^{HW \times HW}$ is large in size, and each element requires numerous multiplications, resulting in high computational cost. The most straightforward way to compute $L$ is to calculate each element $L_{u,v}$ individually. Hence, the total complexity of computing matrix $L$ is $N^2$ times the complexity of computing each element $L_{u,v}$. As defined in Eq. (16) and Eq. (14), $L_{u,v} = \mathcal{L}_{i,j,k,l} = \alpha_{i,j:l}\beta_{i:k,l}$, where $i = \lfloor u/W \rfloor + 1$, $j = u \bmod W$ and $k = \lfloor v/W \rfloor + 1$, $l = v \bmod W$. There are $|l - j|$ and $|k - i|$ multiplication operations in $\alpha_{i,j:l}$ and $\beta_{i:k,l}$, respectively. Here, $i, k$ range from 1 to $H$, and $j, l$ range from 1 to $W$. Therefore, the complexity of $L_{u,v}$ is $\mathcal{O}(N^{\frac{1}{2}})$. Consequently, the overall complex of directly computing $L$ is $\mathcal{O}(N^{\frac{5}{2}})$.

**Efficient Computation.** The polyline path mask $L$ satisfies the conditions in Theorem 1 with $[A^i]_{j,l} = \alpha_{k,j:l}$ and $[B^l]_{i,k} = \beta_{i:k,j}$. Thus, based on Theorem 1, the polyline path mask $L$ can be decomposed as:

$$
L = L^H \times L^V = \hat{L}^H \odot \hat{L}^V.
\tag{30}
$$

For example, as illustrated in Fig. 10 (a), the polyline path mask $L$ in Eq. (16) can be decomposed as:

$$
\tag{31}
$$

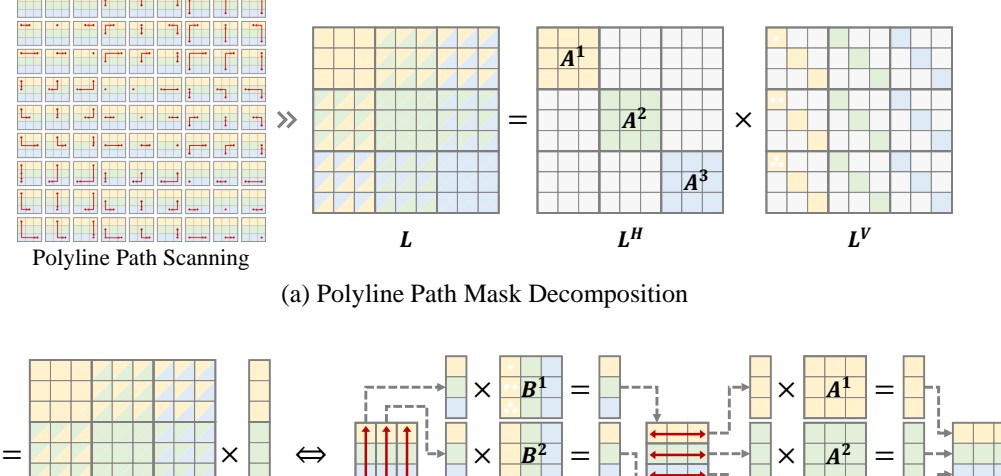

(a) Polyline Path Mask Decomposition

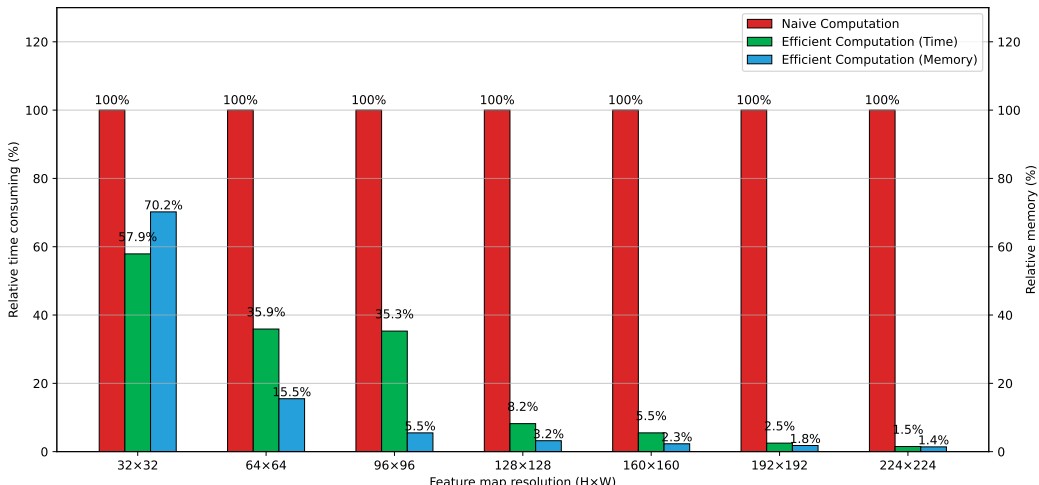

(b) Polyline Path Mask Multiplication

Figure 10: (a) Illustration of the decomposition of the polyline path mask $L$. (b) Illustration of the multiplication between the polyline path mask $L$ and vector $x$. (Algorithm 2) .

Figure 11: The comparison of the relative time consuming and memory usage between the naive computation and efficient computation (Algorithm 2) of $Lx$.

Note that there are $H \times W^2$ non-zero elements in $L^H$ and each non-zero element $\alpha_{i,j:l}$ requires $\mathcal{O}(N^{\frac{1}{2}})$ multiplication operations. Thus, the complexity of computing $L^H$ and $L^V$ is $\mathcal{O}(N^2)$. Similarly, the complexity of computing $\hat{L}^H$ and $\hat{L}^V$ is also $\mathcal{O}(N^2)$. Thus, the complexity of computing $L$ can be reduced to $\mathcal{O}(N^2)$ by Eq. (30).

## A.6 Complexity Analysis of Polyline Path Mask Multiplication

In this section, we analyze the complexity of computing the matrix multiplication between the polyline path mask $L$ and the vector $x$, as stated in Corollary 2 and Algorithm 2, with detailed explanation. Fig. 11 presents the comparison of speed and memory usage between the naive computation and efficient computation of $Lx$. Compared to the naive computation approach, Algorithm 2 achieves substantial speed-up and significantly reduced GPU memory consumption, especially when the shape of $L$ and $x$ is large.

**Corollary 2** (Masked Attention Complexity). *The computational complexity of the matrix multiplication between polyline path mask and vector $\boldsymbol{x}$, i.e., $\boldsymbol{y} = \boldsymbol{L}\boldsymbol{x}$, can be reduced from $\mathcal{O}(N^2)$ to $\mathcal{O}(N^{\frac{3}{2}})$ by Algorithm 2, and further reduced to $\mathcal{O}(N)$ by applying the chunkwise algorithm of Mamba2 [6] to steps 3 and 5 in Algorithm 2.*

**Naive Computation.** Typically, the polyline path mask $\boldsymbol{L} \in \mathbb{R}^{N \times N}$ is a rank-N matrix. Thus, the most direct approach to compute $\boldsymbol{L}\boldsymbol{x}$ requires a computational complexity of $\mathcal{O}(N^2)$.

**Efficient Computation.** As mentioned above, the polyline path mask $\boldsymbol{L}$ can be decomposed as $\boldsymbol{L}^H \times \boldsymbol{L}^V$, where $\boldsymbol{L}^H$ and $\boldsymbol{L}^V$ satisfy the definition in Eq. (18) with $[\boldsymbol{A}^i]_{j,l} = \alpha_{i,j:l}$ and $[\boldsymbol{B}^l]_{i,k} = \beta_{i:k,l}$. Thus, based on Theorem 2, we can design Algorithm 2 for computing the matrix multiplication between polyline path mask $\boldsymbol{L}$ and the vector $\boldsymbol{x}$. As shown in Algorithm 2, computing $\boldsymbol{B}^l \times \boldsymbol{X}_{:,l}$ has a complexity of $\mathcal{O}(H^2)$. Thus, the complexity of computing $\boldsymbol{Z}$ (i.e., step 3 in Algorithm 2) is $\mathcal{O}(H^2W)$. Similarly, the complexity of computing $\boldsymbol{Y}$ (i.e., step 5 in Algorithm 2) is $\mathcal{O}(HW^2)$. Thus, the computational complexity of Algorithm 2 is $\mathcal{O}(N^{\frac{3}{2}})$, where $N = H \times W$.

As illustrated in Fig. 10 (b), the matrices $\boldsymbol{A}^i$ and $\boldsymbol{B}^l$ are symmetric matrices, and their lower triangular parts are both 1-semiseparable matrices as defined in Mamba2 [6]. Therefore, by applying Lemma 1 the complexity of computing $\boldsymbol{B}^l \times \boldsymbol{X}_{:,l}$ and can be reduced from $\mathcal{O}(H^2)$ to $\mathcal{O}(H)$, and the complexity of

---

**Algorithm 2:** Efficient Masked Attention Computation.

**Input:** decay factors $\alpha, \beta$ of the polyline path mask $\boldsymbol{L}$, vector $\boldsymbol{x} \in \mathbb{R}^{HW}$;
1: Compute $\boldsymbol{X} = \text{unvec}(\boldsymbol{x}) \in \mathbb{R}^{H \times W}$;
2: Compute $\boldsymbol{B}^l \in \mathbb{R}^{H \times H}$, where for $l = 1:W$, $[\boldsymbol{B}^l]_{i,k} = \beta_{i:k,l}$;
3: Compute $\boldsymbol{Z} \in \mathbb{R}^{H \times W}$, where $\boldsymbol{Z}_{:,l} = \boldsymbol{B}^l \times \boldsymbol{X}_{:,l}$;
4: Compute $\boldsymbol{A}^i \in \mathbb{R}^{W \times W}$, where for $i = 1:H$, $[\boldsymbol{A}^i]_{j,l} = \alpha_{i,j:l}$;
5: Compute $\boldsymbol{Y} \in \mathbb{R}^{H \times W}$, where $\boldsymbol{Y}_{i,:} = \boldsymbol{A}^i \times \boldsymbol{Z}_{i,:}$;
**Output:** $\boldsymbol{y} = \text{vec}(\boldsymbol{Y})$;

---

computing $\boldsymbol{A}^i \times \boldsymbol{Z}_{i,:}$ can be reduced from $\mathcal{O}(W^2)$ to $\mathcal{O}(W)$. Consequently, the overall complexity of computing $\boldsymbol{L}\boldsymbol{x}$ is $\mathcal{O}(N)$.

### A.7 Applications of Polyline Path Masked Attention

The proposed polyline path mask can be seamlessly integrated into various attention variants in a plug-and-play manner. As illustrated in Fig. 9, theorems and algorithm given in Sec. A.5 and Sec. A.6 ensure that integrating the polyline path mask does not substantially increase the computational complexity of the original attention mechanism. In this section, we introduce several Polyline Path Masked Attention (PPMA), including Polyline Path Masked Vanilla Attention (PPMVA), Polyline Path Masked Linear Attention (PPMLA), Polyline Path Masked Criss-Cross Attention (PPMCCA), and Polyline Path Masked Decomposed Attention (PPMDA).

**Basic Paradigm.** The basic Polyline Path Masked Attention (PPMA) is implemented by performing a Hadamard multiplication with the attention map. Specifically, given query $\boldsymbol{Q}$, key $\boldsymbol{K}$, and value $\boldsymbol{V} \in \mathbb{R}^{HW \times C}$, PPMA is formulated as:

$$
\begin{aligned}
\text{PPMA}\,(\boldsymbol{X}) &= \left(\text{Attn}(\boldsymbol{Q}, \boldsymbol{K}) \odot \boldsymbol{L}^{2D}\right) \boldsymbol{V} \\
&= \left(\text{Attn}(\boldsymbol{Q}, \boldsymbol{K}) \odot \boldsymbol{L}\right) \boldsymbol{V} + \left(\text{Attn}(\boldsymbol{Q}, \boldsymbol{K}) \odot \tilde{\boldsymbol{L}}\right) \boldsymbol{V}.
\end{aligned}
\tag{32}
$$

**1) Polyline Path Masked Vanilla Attention.** According to Eq. (32), the polyline path masked vanilla attention is formulated as:

$$
\text{PPMVA}\,(\boldsymbol{X}) = \left(\text{softmax}(\boldsymbol{Q}\boldsymbol{K}^\top) \odot \boldsymbol{L}^{2D}\right) \boldsymbol{V},
\tag{33}
$$

Based on Corollary 1, the computation of $\boldsymbol{L}^{2D}$ has a complexity of $\mathcal{O}(N^2)$. Thus, Eq. (33) maintains the complexity of $\mathcal{O}(N^2)$.

**2) Polyline Path Masked Linear Attention.** Similar to Mamba2's attention form (Eq. (28)), the polyline path masked linear attention is formulated as:

$$
\text{PPMLA}\,(\boldsymbol{X}) = \left((\boldsymbol{Q}\boldsymbol{K}^\top) \odot \boldsymbol{L}^{2D}\right) \boldsymbol{V} = \left((\boldsymbol{Q}\boldsymbol{K}^\top) \odot \boldsymbol{L}\right) \boldsymbol{V} + \left((\boldsymbol{Q}\boldsymbol{K}^\top) \odot \tilde{\boldsymbol{L}}\right) \boldsymbol{V}.
\tag{34}
$$

Based on Theorem 3, we can compute $\left((\boldsymbol{Q}\boldsymbol{K}^\top)\odot\boldsymbol{L}\right)\boldsymbol{V}$ as follows:

$$\boldsymbol{\mathcal{K}}^V = \texttt{einsum}\left('\texttt{md},\texttt{mc}\to\texttt{mdc}',\boldsymbol{K},\boldsymbol{V}\right)$$
$$\boldsymbol{\mathcal{L}}^{KV} = \texttt{einsum}\left('\texttt{nm},\texttt{mdc}\to\texttt{mdc}',\boldsymbol{L},\boldsymbol{\mathcal{K}}^V\right) \tag{35}$$
$$\boldsymbol{Y} = \texttt{einsum}\left('\texttt{md},\texttt{mdc}\to\texttt{mc}',\boldsymbol{Q},\boldsymbol{\mathcal{L}}^{KV}\right).$$

Eq. (35) shows that the computational complexity of Eq. (34) depends on computing $\boldsymbol{\mathcal{L}}^{KV}$. Based on Corollary 2 and Algorithm 2, the computational complexity of the second line in Eq. (35) can be reduced from $\mathcal{O}(N^2)$ to $\mathcal{O}(N)$. Thus, the computational complexity of Eq. (34) maintains $\mathcal{O}(N)$.

**3) Polyline Path Masked Criss-Cross Attention.** The original criss-cross attention [22] employs sparse attention over tokens located in the same row or column, achieving a computational complexity of $\mathcal{O}(N^{\frac{3}{2}})$. In this work, following RMT [10], we decompose criss-cross attention into vertical attention over each column followed by horizontal attention over each row. The polyline path masked criss-cross attention is formulated as:

$$\mathrm{PPMCCA}\left(\boldsymbol{X}\right) = \left((\boldsymbol{S}^H\times\boldsymbol{S}^V)\odot\boldsymbol{L}^{2D}\right)\boldsymbol{V} = \left((\boldsymbol{S}^H\times\boldsymbol{S}^V)\odot\boldsymbol{L}\right)\boldsymbol{V} + \left((\boldsymbol{S}^H\times\boldsymbol{S}^V)\odot\tilde{\boldsymbol{L}}\right)\boldsymbol{V}, \tag{36}$$

where horizontal and vertical attention maps $\boldsymbol{S}^H, \boldsymbol{S}^V \in \mathbb{R}^{HW\times HW}$ satisfy the form in Eq. (18) with $\boldsymbol{A}^i = \mathrm{softmax}(\boldsymbol{\mathcal{Q}}_{i,:,:}\boldsymbol{\mathcal{K}}_{i,:,:}^\top)$ and $\boldsymbol{B}^l = \mathrm{softmax}(\boldsymbol{\mathcal{Q}}_{:,l,:}\boldsymbol{\mathcal{K}}_{:,l,:}^\top)$, and $\boldsymbol{\mathcal{Q}}, \boldsymbol{\mathcal{K}} \in \mathbb{R}^{H\times W\times C}$ are tensor forms of $\boldsymbol{Q}, \boldsymbol{K}$, respectively [22]. Based on Theorem 1, we can reformulate the left part of Eq. (36) as:

$$\begin{aligned}
\left((\boldsymbol{S}^H\times\boldsymbol{S}^V)\odot\boldsymbol{L}\right)\boldsymbol{V} &= \left((\boldsymbol{S}^H\times\boldsymbol{S}^V)\odot(\boldsymbol{L}^H\times\boldsymbol{L}^V)\right)\boldsymbol{V}\\
&= \left(\left(\hat{\boldsymbol{S}}^H\odot\hat{\boldsymbol{S}}^V\right)\odot\left(\hat{\boldsymbol{L}}^H\odot\hat{\boldsymbol{L}}^V\right)\right)\boldsymbol{V}\\
&= \left(\left(\hat{\boldsymbol{S}}^H\odot\hat{\boldsymbol{L}}^H\right)\odot\left(\hat{\boldsymbol{S}}^V\odot\hat{\boldsymbol{L}}^V\right)\right)\boldsymbol{V} \qquad (37)\\
&= \left(\boldsymbol{S}^H\odot\boldsymbol{L}^H\right)\times\left(\boldsymbol{S}^V\odot\boldsymbol{L}^V\right)\times\boldsymbol{V}\\
&= \left(\boldsymbol{S}^H\odot\boldsymbol{L}^H\right)\times\left(\left(\boldsymbol{S}^V\odot\boldsymbol{L}^V\right)\times\boldsymbol{V}\right).
\end{aligned}$$

Note that matrices $\hat{\boldsymbol{S}}^H\odot\hat{\boldsymbol{L}}^H$ and $\hat{\boldsymbol{S}}^V\odot\hat{\boldsymbol{L}}^V$ also satisfy the form (i.e. $\boldsymbol{M}^A$ and $\boldsymbol{M}^B$) in Eq. (18). Thus, the computational complexity of Eq. (37) can be reduced to $\mathcal{O}(N^{\frac{3}{2}})$ by Algorithm 2. Similar conclusions can also be derived for the right part of Eq. (36). Thus, the overall computational complexity of Eq.(36) maintains $\mathcal{O}(N^{\frac{3}{2}})$.

**4) Polyline Path Masked Decomposed Attention.** For general decomposable attention which can be decomposed as $\boldsymbol{S} = \boldsymbol{S}_1\times\boldsymbol{S}_2$, where $\boldsymbol{S}_1\in\mathbb{R}^{N\times D}$ and $\boldsymbol{S}_2\in\mathbb{R}^{D\times N}$, the polyline path masked decomposed attention is formulated as:

$$\mathrm{PPDA}\left(\boldsymbol{X}\right) = \left((\boldsymbol{S}_1\times\boldsymbol{S}_2)\odot\boldsymbol{L}^{2D}\right)\boldsymbol{V} = \left((\boldsymbol{S}_1\times\boldsymbol{S}_2)\odot\boldsymbol{L}\right)\boldsymbol{V} + \left((\boldsymbol{S}_1\times\boldsymbol{S}_2)\odot\tilde{\boldsymbol{L}}\right)\boldsymbol{V} \tag{38}$$

According to Theorem 3, we can compute $\left((\boldsymbol{S}_1\times\boldsymbol{S}_2)\odot\boldsymbol{L}\right)\boldsymbol{V}$ as follows:

$$\boldsymbol{\mathcal{S}}_2^V = \texttt{einsum}\left('\texttt{md},\texttt{mc}\to\texttt{mdc}',\boldsymbol{S}_2^\top,\boldsymbol{V}\right)$$
$$\boldsymbol{\mathcal{L}}^{SV} = \texttt{einsum}\left('\texttt{nm},\texttt{mdc}\to\texttt{mdc}',\boldsymbol{L},\boldsymbol{\mathcal{S}}_2^V\right) \tag{39}$$
$$\boldsymbol{Y} = \texttt{einsum}\left('\texttt{md},\texttt{mdc}\to\texttt{mc}',\boldsymbol{S}_1,\boldsymbol{\mathcal{L}}^{SV}\right).$$

Based on Corollary 2 and Algorithm 2, the computational complexity of Eq. (39) can be reduced from $\mathcal{O}(N^2)$ to $\mathcal{O}(ND)$. Thus, the computational complexity of Eq. (38) is $\mathcal{O}(ND)$.

## B  Experimental Details

### B.1  Architecture Details

As illustrated in Fig. 5, our backbone adopts the same four-stage hierarchical architecture as RMT [10], where the first three stages employ Polyline Path Masked Criss-Cross Attention and the final stage

employs the Polyline Path Masked Vanilla Attention. Moreover, we develop our model in three scales: tiny (PPMA-T), small (PPMA-S), and base (PPMA-B).

The detailed configurations of PPMA variants are provided in Tab. 5. Following RMT [10], the stem layer consists of five $3 \times 3$ convolution layers followed by GELU and batch normalization to embed the input image into $56 \times 56$ tokens. The downsampling layer consists of $3 \times 3$ convolution layers with stride 2 to reduce the feature map's resolution. Moreover, we follow RMT [10] and incorporate RoPE [38], CPE [3], and LCE [57] into the PPMA blocks. All other configurations also follow RMT [10]. Code is available at `https://github.com/zhongchenzhao/PPMA`.

## B.2   Training Settings for ImageNet-1K

To ensure reproducibility and consistency with prior work, we follow the training strategy of RMT [10] and DeiT [44]. Specifically, we employ various data augmentation techniques, including RandAugment [5], Mixup [52] (prob=0.8), CutMix [50] (prob=1.0), Random Erasing [55] (prob=0.25). For model optimization, we use the AdamW optimizer with a cosine decay learning rate scheduler and train our model 300 epochs from scratch. The initial learning rate, weight decay, and batch size are set to 0.001, 0.05, and 1024, respectively. The drop path rates for PPMA-T, PPMA-S, and PPMA-B are set to 0.1, 0.15, and 0.4, respectively. We also adopt training techniques from RMT [10], including Label Smoothing (0.1) [41] and Exponential Moving Average (EMA) [33].

## B.3   Training Settings for Downstream Tasks

For experiments on the ADE20K [56] and MSCOCO2017 [28] datasets, we follow the training settings of TransNeXT [35], and utilize the MMDetection [2] and MMSegmentation [4] libraries for training. Specifically, in the MMDetection [2] library, we adopt Mask R-CNN [18] as the basic framework and use the AdamW optimizer with an initial learning rate of $0.0001$ and a weight decay of $0.01$. The model is trained for 12 epochs with a batch size of 16 using the standard $1\times$ schedule. In the MMSegmentation [4] library, we adopt UPerNet [48] as the basic framework and use the AdamW optimizer with the initial learning rate of $6 \times 10^{-5}$ and the weight decay of 0.01. All models are trained for 160K iterations with a batch size of 16 on the ADE20K dataset. The input size of images is set to $512 \times 512$ .

## B.4   Throughput Comparison

To evaluate the inference speed of our model, we measure the throughput of PPMA-T/S/B on an A800 GPU with a batch size of 64 and the image resolution of $224 \times 224$. As shown in Table 6, the inference throughput of PPMA-T/S/B decrease by 37%/30%/21% compared to RMT-T/S/B, respectively. This is mainly caused by the additional GPU kernel launches and memory transactions required to compute the polyline path mask. As shown in Table 6, the CUDA implementation of TransNeXt achieves a significant speedup over the PyTorch implementation. In our implementation, the polyline path mask is currently computed using PyTorch. Similar to TransNeXt, our implementation can also be optimized through engineering efforts, such as using CUDA or Triton-based implementations, to accelerate inference speed.

## B.5   Visualization

The visualizations of the polyline path masked attention map are shown in Fig. 12. Input images are taken from the ImageNet-1K validation set, and the query token is marked by a red box on each input

| Model | Blocks | Channels | Heads | Ratios | #Param. (M) | FLOPs (G) |
|---|---|---|---|---|---|---|
| PPMA-T | [2, 2, 8, 2] | [64, 128, 256, 512] | [4, 4, 8, 16] | [3, 3, 3, 3] | 14 | 2.7 |
| PPMA-S | [3, 4, 18, 4] | [64, 128, 256, 512] | [4, 4, 8, 16] | [4, 4, 3, 3] | 27 | 4.9 |
| PPMA-B | [4, 8, 25, 8] | [80, 160, 320, 512] | [5, 5, 10, 16] | [4, 4, 3, 3] | 54 | 10.6 |

Table 5: Detailed Architectures of the Polyline Path Masked Attention based Vision Transformer.

Table 6: Comparison of inference speed across different models on ImageNet-1K. Throughput is measured on an A800 GPU with a batch size of 64.

| Model | #Param. (M) | FLOPs (G) | Throughput (imgs/s) | Top-1 (%) |
|---|---|---|---|---|
| BiFormer-T [57] | 13 | 2.2 | 1602 | 81.4 |
| SMT-T [29] | 12 | 2.4 | 636 | 82.2 |
| RMT-T [10] | 14 | 2.5 | 1650 | 82.4 |
| TransNeXt-M (PyTorch) [35] | 13 | 2.7 | 742 | 82.5 |
| TransNeXt-M (CUDA) [35] | 13 | 2.7 | 1299 | 82.5 |
| PPMA-T | 14 | 2.7 | 1034 | 82.6 |
| CMT-S [14] | 25 | 4.0 | 848 | 83.5 |
| MaxViT-T [45] | 31 | 5.6 | 826 | 83.6 |
| SMT-S [29] | 20 | 4.8 | 356 | 83.7 |
| BiFormer-S [57] | 26 | 4.5 | 766 | 83.8 |
| RMT-S [10] | 27 | 4.5 | 876 | 84.0 |
| TransNeXt-T (PyTorch) [35] | 28 | 5.7 | 508 | 84.0 |
| TransNeXt-T (CUDA) [35] | 28 | 5.7 | 947 | 84.0 |
| PPMA-S | 27 | 4.9 | 612 | 84.2 |
| SMT-B [29] | 32 | 7.7 | 237 | 84.3 |
| BiFormer-B [57] | 57 | 9.8 | 498 | 84.3 |
| CMT-B [14] | 46 | 9.3 | 447 | 84.5 |
| TransNeXt-T (PyTorch) [35] | 50 | 10.3 | 266 | 84.7 |
| TransNeXt-T (CUDA) [35] | 50 | 10.3 | 436 | 84.7 |
| RMT-B [10] | 54 | 9.7 | 457 | 84.9 |
| PPMA-B | 54 | 10.6 | 362 | 85.0 |

image. The decay factors and attention maps are generated by the second block of the first stage in the PPMA-T model trained on the ImageNet-1K training set.

**Input-dependent Decay Factor.** As shown in Fig. 12 (b) and (c), the decay factors $\alpha$ and $\beta$ learned by the network can roughly capture the edge information of objects in the feature map: decay factors at edges tend to be smaller (approaching zero), whereas those in homogeneous regions tend to be larger (approaching one). Moreover, the supervised training encourages the decay factors $\alpha$ and $\beta$ to focus on horizontal and vertical edge information, respectively.

**Polyline Path Mask.** As shown in Fig. 12 (e), the polyline path mask, generated by the cumulative multiplication of decay factors, effectively captures the semantic continuity in the feature space. It maintains continuity in homogeneous regions sharing the same semantics and shows discontinuity at the edges between regions of different semantics.

**Polyline Path Masked Attention Map.** Fig. 12 (d) shows that attention maps from shallow layers in typical ViT models often struggle to focus on tokens relevant to the query token. In contrast, Fig. 12 (f) demonstrates that integrating the polyline path mask $\boldsymbol{L}^{2D}$ successfully suppresses interference from distant and irrelevant tokens, resulting in more semantically accurate masked attention maps.

## C  Discussion

### C.1  Selectivity of Polyline Path Mask

Compared to previous state-space models (SSMs) such as RetNet [40], the primary contribution of Mamba [11] and Mamba2 [6] is the introduction of a selective mechanism into the structured mask, which leads to significant performance improvements. However, current studies still lack a deep understanding of this crucial selectivity mechanism.

In this work, we argue that the selective mechanism in Mamba explicitly models the semantic continuity in sequences, which corresponds to the local smoothness prior in images. Building on this insight, we adopt the selective structured mask of Mamba2 and naturally generalize it into a 2D polyline path mask for Vision Transformers (ViTs).

**Semantic Continuity in Sequence.** Similar to self-attention maps in ViTs, the structured mask $\boldsymbol{L} \in \mathbb{R}^{N \times N}$ can also be viewed as a weighting matrix that maps input tokens $\boldsymbol{X} \in \mathbb{R}^{N \times C}$ to output

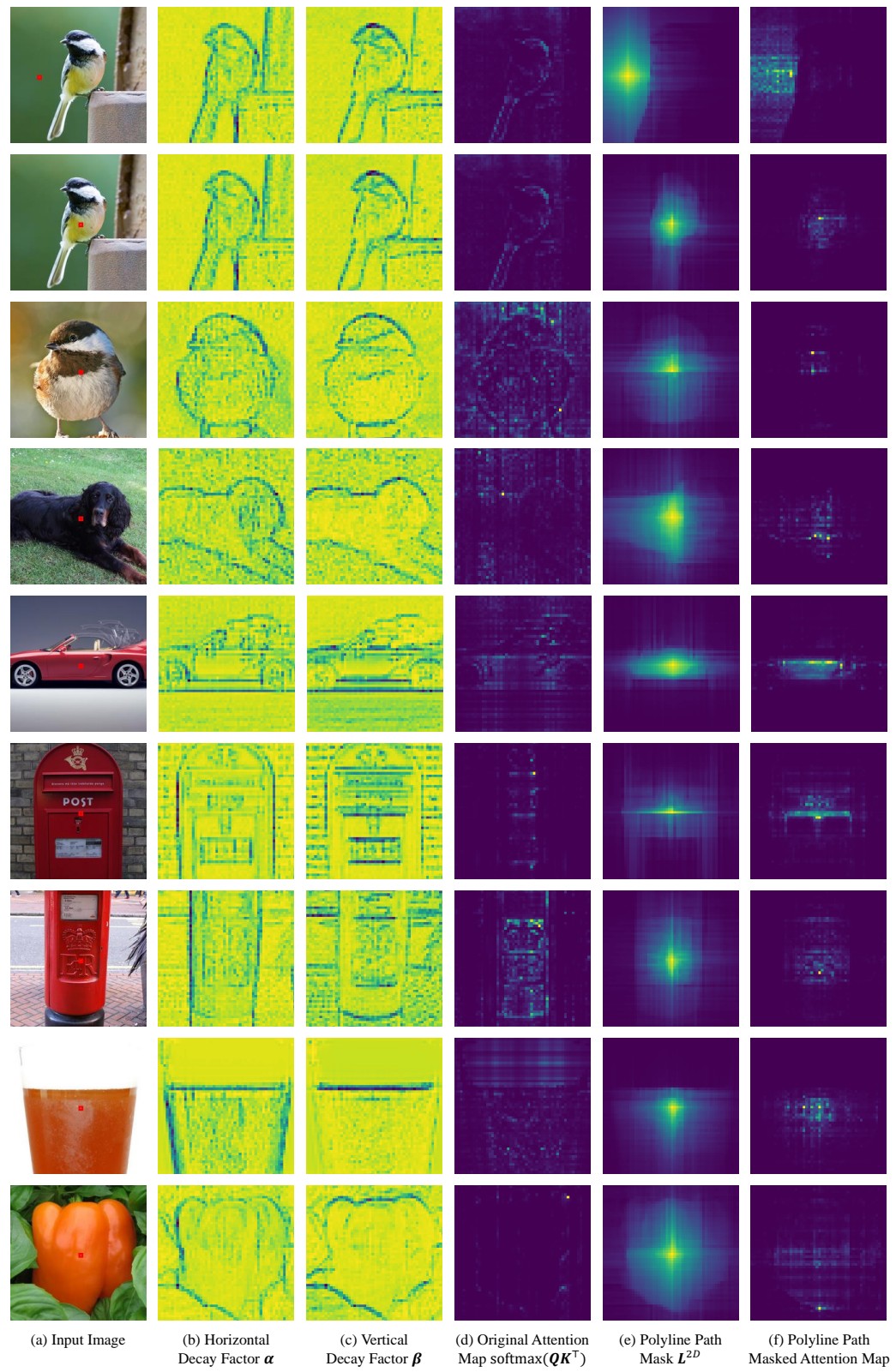

| (a) Input Image | (b) Horizontal Decay Factor $\boldsymbol{\alpha}$ | (c) Vertical Decay Factor $\boldsymbol{\beta}$ | (d) Original Attention Map softmax($\boldsymbol{QK}^\top$) | (e) Polyline Path Mask $\boldsymbol{L}^{2D}$ | (f) Polyline Path Masked Attention Map |

Figure 12: Visualizations of the decay factors and the polyline path masked attention maps of the well-trained PPMA model. In each input image, the query token is marked by a red box.

tokens $\boldsymbol{Y} \in \mathbb{R}^{N \times C}$ along the sequence length dimension. In this weighting matrix $\boldsymbol{L}$, a larger decay weight $\boldsymbol{L}_{i,j}$ indicates a greater influence of the input token $\boldsymbol{X}_i$ on the output token $\boldsymbol{Y}_j$, and vice versa. In Mamba2, $\boldsymbol{L}_{i,j}$ is computed as the cumulative multiplication of decay factors $a_{i:j}^{\times}$ to achieve linear complexity. As a result, if any factor is close to zero, $\boldsymbol{L}_{i,j}$ approaches zero; conversely, $\boldsymbol{L}_{i,j}$ approaches one only when all decay factors are close to one.

For most semantic-related tasks, an ideal structured mask should model semantic continuity in the sequence: it should maintain continuous between connected tokens with the same semantic, while breaking between tokens with different semantics. This enables the aggregation and separation of tokens according to their semantics. Accordingly, decay factors should ideally be larger in homogeneous regions and smaller at heterogeneous regions. As illustrated in Fig. 12 (b) and (c), the decay factors learned through supervised training align well with this assumption.

**Local Smoothness Prior in Images.** In natural images, spatially adjacent patches are more likely to belong to the same object and share similar semantics. This local smoothness prior plays a crucial role in natural image processing tasks, especially those requiring fine-grained feature extraction. The selectivity of polyline path mask aligns naturally with this prior by modeling semantic continuity within homogeneous regions and allowing discontinuities at object edges. Experimental results also show that integrating the polyline path mask yields significant performance improvements on the ADE20K semantic segmentation task.

## C.2   3D Extension of Polyline Path Mask

Based on the decomposability, we naturally extend the 2D poly-line path mask to 3D applications. As illustrated in Fig. 13, the 3D polyline path mask $\boldsymbol{L}^{3D}$ can be decomposed as the multiplication of three 1D structured masks, $\boldsymbol{L}^H \times \boldsymbol{L}^V \times \boldsymbol{L}^D$, representing the horizontal, vertical, and depth scanning masks, respectively. Specifically, for each token pair $(\boldsymbol{x}_{i,j,k}, \boldsymbol{x}_{l,m,n})$ in the 3D grid, the 3D polyline path mask is defined as:

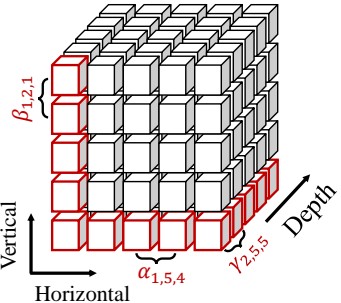

$$\mathcal{L}^{3D}_{(i,j,k),(l,m,n)} = \alpha_{i,j,k:n}\beta_{i,j:m,n}\gamma_{i:l,m,n}, \tag{40}$$

where $\mathcal{L}^{3D}$ is the tensor form of matrix $\boldsymbol{L}^{3D}$, $\alpha$, $\beta$, and $\gamma$ are the decay factors along the horizontal, vertical, and depth axes, respectively. Compared to the cross-scanning strategy [30], the 3D polyline path scanning strategy better preserves the adjacency relationships of 3D tokens.

Figure 13: Illustration of the 3D extension of polyline path mask.

## C.3   Limitations

In this work, we introduce a learnable, input-dependent polyline path mask as the explicit positional encoding for ViTs, replacing the fixed decay mask in RMT [10]. Experiments on high-level tasks demonstrate the superiority of our method, particularly in fine-grained segmentation benchmarks, where PPMA-T/S/B outperform RMT-T/S/B by 0.7%/1.3%/0.3% SS mIoU on ADE20K, respectively.

Notably, our carefully designed polyline path mask $\boldsymbol{L}^{2D}$ is decomposable as described in Eq. (30), enabling efficient computation via algorithms 2 to optimize the computational complexity. However, despite these optimizations, the large size of the mask $\boldsymbol{L}^{2D}$ still inevitably incurs extra GPU memory occupation and slower inference speed compared to RMT [10]. As shown in Table 6, the inference throughput of PPMA-T/S/B decrease by 37%/30%/21% in comparison with RMT-T/S/B, respectively. This limitation can be mitigated through engineering optimizations, such as CUDA or Triton-based implementations, which we plan to investigate in the future work.

