# OpenReview forum: "Polyline Path Masked Attention for Vision Transformer"
_NeurIPS.cc/2025/Conference — NeurIPS 2025 spotlight_

### Official Review · Reviewer_SWqC · 2025-06-30

**Clarity:** 2
**Significance:** 3
**Originality:** 3
**Rating:** 4
**Confidence:** 3

**Summary:**

This paper proposes a new relative positional encoding, called polyline path scanning/mask, for ViT, where the adjacency of visual tokens in 2d images can be maintained under the Manhattan distance.

**Questions:**

The paper starts with a discussion on ViT and Mamba (RNN) for CV tasks, and how ViT and Mamba2 can be more effectively combined. However, it turns out later that this is NOT the focus of the research at all.

The proposed relative position encoding method is orthogonal to Mamba, and can be used in any models that require to encoding position info of visual tokens. The detailed discussion of pros and cons of Mamba2 and Mamba-ViT hybrid models seem unnecessary. Since Mamba-ViT hybrid models have been comprehensively explored in language models, such a discussion seems dilute the novelty of the paper. The paper should have focused on positional encoding for visual tokens, using Mamba2 / VIT as justification and experimentation.

**Ethical Concerns:**

["NO or VERY MINOR ethics concerns only"]

**Limitations:**

yes

**Quality:**

3

**Strengths And Weaknesses:**

Strengths: The new positional encoding method, as detailed in Sec. 4, is well-motivated, simple to implemented, and mathematically sound. The effectiveness of well-justified in the empirical study.

Weaknesses: the presentation can be improved. The paper starts with a discussion on ViT and Mamba (RNN) for CV tasks, and how ViT and Mamba2 can be more effectively combined. However, it turns out later that this is NOT the focus of the research at all. The proposed relative position encoding method is orthogonal to Mamba, and can be used in any models that require to encoding position info of visual tokens. The detailed discussion of pros and cons of Mamba2 and Mamba-ViT hybrid models seem unnecessary. Since Mamba-ViT hybrid models have been comprehensively explored in language models, such a discussion seems dilute the novelty of the paper. The paper should have focused on positional encoding for visual tokens, using Mamba2 / VIT as justification and experimentation.

---

> ### Author Rebuttal · Authors · 2025-07-30
>
> Thank you very much for your positive feedback. We are delighted that you consider our method to be “**well-motivated, simple to implement, and mathematically sound**”. Please see below for our point-by-point responses to your comments.
>
>
>
> - **Questions 1**: The presentation can be improved. The paper starts with a discussion on ViT and Mamba (RNN) for CV tasks, and how ViT and Mamba2 can be more effectively combined. However, it turns out later that this is NOT the focus of the research at all.
>
>   > **Answer**: Thank you for your valuable feedback. We acknowledge that the initial presentation may have caused some confusion, and we will **revise and reorganize the introduction and motivation** in the final version to more clearly convey the core focus of our work.
>   >
>   > We would also like to clarify that, while our core contribution lies in the proposed polyline path mask $L^{2D}$, the motivation of our study is indeed to **leverage the complementary strengths of Mamba2 and ViTs**. Furthermore, the proposed method is not directly constructed by human intuition alone, but is naturally derived through refinements within the Mamba2 framework. Frankly speaking, the polyline path mask can be seen as **a carefully adapted and extended 2D variant** of the structured mask $L^{1D}$ in Mamba2, tailored for image data.
>
>
>
> - **Questions 2**: The proposed relative position encoding method is orthogonal to Mamba, and can be used in any models that require encoding position info of visual tokens. The detailed discussion of pros and cons of Mamba2 and Mamba-ViT hybrid models seems unnecessary. Since Mamba-ViT hybrid models have been comprehensively explored in language models, such a discussion seems to dilute the novelty of the paper. The paper should have focused on positional encoding for visual tokens, using Mamba2 / ViT as justification and experimentation.
>
>   > **Answer**: Thank you very much for your valuable comments. We appreciate your suggestion and will revise the manuscript to clarify the novelty and focus of our work. We agree that the proposed mask, as a form of relative positional encoding, can be applied to various attention mechanisms beyond Mamba2.
>   >
>   > We also agree that **the core novelty of our work lies in the design of the 2D polyline path mask**. In Section 4, we provide a systematic exposition of this design, including its **definition**, **properties** (e.g., decomposability), **theoretical guarantees** (efficient algorithm), and **practical applications** (e.g., PPMVA and PPMCCA).
>   >
>   > In the final version, we will further refine the structure of the *Introduction* and *Methodology* sections to place stronger emphasis on the proposed mask as a general-purpose positional encoding method for visual tokens.

---

### Official Review · Reviewer_gCJq · 2025-07-02

**Clarity:** 3
**Significance:** 3
**Originality:** 3
**Rating:** 5
**Confidence:** 3

**Summary:**

The authors propose a new way of casting attention by incorporating a new structures mask inspired by the Mamba2 architectures into classical self-attention – and demonstrate that this explicit modelling of the spatial adjacency priors present in images can improve results across various vision tasks.

**Questions:**

**[Q1]:**  RMT-results repeatedly stated lower than in original RMT paper: The results presented in this work for RMT on various tasks is consistently lower than what is stated in the original RMT paper (both arxiv and CVPR version); I’d like the authors to comment on this, as the original RMT results are mostly on par and sometimes better than the results of the method proposed here!
Examples:
$\textrightarrow$  Detection: RMT-T stated as 46.7 instead of 47.1 (APb); 68.6 vs 68.8 (APb50); 51.6 vs 51.7 (APb75) | 42.1 vs 42.6 (APm), 65.3 vs 65.8 (APm50), 45.2 vs 45.9 (APm75).  Similar for other model sizes.
$\textrightarrow$  Classification: RMT-B stated as 84.9 instead of 85.0 (orig.), RMT-S as 84.0 instead of 84.1 (orig.)

*Note*: I don’t expect a new method to outperform all others, but this needs clarification!

**[Q2]:**  Given the claim of ‘plug-and-play’ integration of the mask, I would be quite interested how the method performs on a plain ‘monolithic’ ViT (i.e. no down-sampling, etc);
$\textrightarrow$  Can the masking approach be simply integrated, and if yes, how does it perform?  (I think this could provide quite distinct and ‘clean’ insights, given that ViT is a very simply yet powerful architecture)
$\textrightarrow$  Given that it is inspired by Mamba2, and other Vision-Mamba works exist – have the authors ablated how simply switching out their mask would perform in such architectures?

**[Q3]:**  FLOP counts only provide a partial impression of the actual efficiency of a model; While I can see that some empirical throughput is provided in the appendix (Table 3), it would be interesting to add this to the step-by-step ablation in Table 4 of the main paper to have a side-by-side comparison of benefit vs. throughput decrease!

**[Q4]:** What exactly is the difference of your scanning approach to 2DMamba? I see that some comparison is provided in the appendix within the novelty statement, but this seems to relate mainly to the fact that the entire 2DMamba approach is built on Mamba(1), not the scanning strategy itself. Some clarification/contrasting would be valuable here.

*Minor*:
Table 4 and Figure 6 have a slight mismatch that could be improved: Images show the Hilbert map while the Table does not (and instead V2H); Replacing the ‘w/o Mask’ image through the V2H one and adding the Hilbert-Mask results to the Table would improve consistency.

---
**TL;DR**: I like the paper, and am happy to reconsider my rating once the questions have been addressed -- but especially the first one is an issue that needs clarification!

---
---
## Post-Rebuttal Update:
As detailed below, my questions and concerns have been sufficiently addressed, and I have raised my score accordingly.

**Ethical Concerns:**

["NO or VERY MINOR ethics concerns only"]

**Final Justification:**

The authors have provided **thoughtful responses to all my questions, including new insights and additional results where required**.

I have also read through all the other reviewers' questions, and I did not find any additional ones that would raise concerns.
The authors also provide many valuable insights in the appendix, and recommendations to include a few into the main paper were made by me and other reviewers.

I'd like to also highlight the authors' **proper and honest discussion** of potential limitations of their method, which I value quite highly (as pointed out in my review already).

**$\rightarrow$ I do not see any prohibitive concerns, and recommend the paper for acceptance.**

**Limitations:**

Limitations sufficiently discussed: The authors briefly discuss some limitations in the conclusion, and add another more detailed discussion section in the appendix – which could be moved to the main paper if space permits in the final version, given that it provides an **honest discussion of actual drawbacks, which I very much appreciate!**

**Paper Formatting Concerns:**

None; Questions in checklist also thoroughly addressed.

**Quality:**

3

**Strengths And Weaknesses:**

## Strengths
**Originality & Significance:**
- Clear motivation of the method based on the identified strengths and weaknesses in Mamba and ViT-like architectures – providing the basis for the approach proposed here
- The authors demonstrate the applicability via improvements across three different vision tasks and for three model sizes

**Quality:**
- Contributions are generally well-placed within the wider area of research
- Quite detailed and rigorous introduction of the polyline path mask computation (including underlying theory how to make it efficient)
- Appropriate number of experiments across tasks to show versatility of method

**Clarity:**
- Contributions and underlying motivation clearly outlined in the introduction
- Explanations well-supported through a good mix of figures, algorithm and formulas
- The paper is well written and easy to read and follow; Further details found in the appendix, but the paper provides a set of great visualizations paired with the main formulas to easily and thoroughly explain the main ideas


## Weaknesses
- Inconsistency between the results presented in the RMT paper and the ones stated here for RMT – stated repeatedly lower (see questions)
- Paper could benefit from integrating the mask in other architectures – given the claim that it works in a ‘plug-and-play’ manner; see questions
- Some smaller inconsistencies in comprehensiveness of results, see questions.

---

> ### Author Rebuttal · Authors · 2025-07-30
>
> We sincerely thank the reviewer for the positive evaluation of our work. We truly appreciate your recognition of the **"clear motivation"**, **"quite detailed and rigorous introduction of the polyline path mask computation"**, and **"well-written" paper paired with "great visualizations"**. We feel very fortunate to receive such a high-quality review. Please see below for our point-by-point responses to your comments.
>
> - **Question 1**: RMT-results repeatedly stated lower than those in original paper.
>   > **Answer**: Thank you for the constructive comment. We can clarify this issue from the following three aspects:
>   >
>   > **1) Discrepancy Caused by Batch Size Differences.**  As the most important competing counterpart, **we reproduced RMT results under the same experimental settings as ours**.  Specifically, both RMT and our PPMA are trained with a batch size of **1024**, due to hardware constraints—we only have access to a single 8-GPU (RTX 4090 with 24 GB memory) machine, which couldn't support a larger batch size.
>   >
>   > **The discrepancies between our reproduced RMT results and those in the original paper stem from batch size differences during training**.  The released RMT weights provided in GitHub repository were trained with a batch size of **2048** on **16 A100 GPUs** (80GB memory). It is worth noting that, possibly due to an oversight, the RMT paper states that a batch size of 1024 was used on ImageNet-1K, which differs from the information disclosed in the GitHub repository. This can be confirmed by the code snippet below:
>   >
>   > ```
>   > checkpoints = vars(torch.load(r"download/RMT-B.pth")['args'])
>   > print(f"gpu_num: {checkpoints['world_size']}, batch_size_per_gpu: {checkpoints['batch_size']}")
>   > print(f"batch size: {checkpoints['world_size']*checkpoints['batch_size']}")
>   > """
>   > >>> gpu_num: 16, batch_size_per_gpu: 128
>   > >>> batch size: 2048
>   > """
>   > # world_size = the actual number of GPUs used
>   > ```
>   >
>   > Batch size can affect classification results on ImageNet-1K by **0.1% to 0.2%**, which also affects downstream COCO detection results since they rely on ImageNet-1K pretrained weights.
>   >
>   > **2) We made sure to reproduce RMT fairly and without bias**.  In fact, we have rigorously verified RMT results via multiple repeated experiments.  For example, RMT-S was trained on ImageNet-1K three times using the official code, achieving Top-1 accuracy of 83.86%, 83.92%, and 83.95%. We reported the best (84.0%) in Table 1. In addition, we have included the comparison results on **MSCOCO 2017** using Mask R-CNN with the **3× MS schedule** (36 epochs with multi-scale inputs), where PPMA-B outperforms RMT-B.
>   > \begin{array}{lcccccc}\hline \text{Backbone}&AP^b &AP^b_{50} &AP^b_{75} &AP^m &AP^m_{50} &AP^m_{75}\newline\hline \text{RMT-B (original paper)}&52.2&72.9&57.0&46.1&\bf{70.4}&49.9\newline \text{RMT-B (reproduction)}& 52.1&73.1&57.2&46.2&70.3&50.0\newline \text{PPMA-B (Ours)}&\bf{52.6}&\bf{73.3}&\bf{57.5}&\bf{46.3}&70.3&\bf{50.2} \newline \hline\end{array}
>   >
>   > **3) Theoretical relationship to RMT.** Importantly, we would like to emphasize that **RMT can be seen as a special case of our method**, where all learnable decay factors are fixed constants. This relationship is analogous to that between Mamba2 and RetNet. Therefore, **our PPMA model is theoretically at least as expressive as RMT**, and potentially more powerful in practice.
>   >
>   > ------
>   >
>   > In conclusion, we are confident that **all experiments in our paper—including RMT baselines—were conducted under fair and reproducible settings**. To ensure transparency, we released the complete training code, conda environments, training logs, and checkpoints for both **PPMA and reproduced RMT** in the anonymous code (hosted on Google Drive) a month ago.
>   >
>   > As suggested, we will explicitly clarify the aforementioned experimental settings in the final version.
>
> - **Question 2.1**: About ‘plug-and-play’ claim: how the method performs on a plain ‘monolithic’ ViT (i.e., no down-sampling)? Can the mask be simply integrated, and if yes, how does it perform?
>   > **Answer**: Yes. The integration process can be easily understood by comparing the formulations of self-attention and PPMA:
>   > $$
>   > {\rm Attention}(x)={\rm softmax}(QK^\top)V.
>   > $$
>   >
>   > $$
>   > {\rm PPMVA}(x)=({\rm softmax}(QK^\top)\odot L^{2D})V.
>   > $$
>   >
>   > Simply performing element-wise multiplication between the attention map and our mask enables seamless integration. Thus, we consider the proposed mask to be 'plug-and-play'.
>   >
>   > **Example on plain ‘monolithic’ ViT:** We use the **DeiT** model (which does not involve down-sampling) as an example. Only a few lines of DeiT code need to be modified:
>   >
>   > ```
>   > mask = self.generate_polyline_path_mask(x)
>   > attn = (q @ k.transpose(-2, -1)) * self.scale
>   >
>   > # attn = attn.softmax(dim=-1) # PPMA just replaces this line with the next line
>   > attn = (0.5*torch.softmax(attn + mask, -1) + 0.5*torch.softmax(attn + mask.permute(0, 1, 3, 2), -1))
>   > y = attn @ v
>   > ```
>   >
>   > Experimental results with and without the polyline path mask on ImageNet-1K are shown in the table below:
>   >
>   > \begin{array}{lccc}\hline \text{Model}&\text{Param. (M)}&\text{FLOPs (G)}&\text{Top-1 Acc.}(\\%) \newline\hline \text{DeiT-tiny}&5&1.3&72.2\newline \text{DeiT-tiny+PPMA}&5&1.3&\bf{72.7} \newline \hline\end{array}
>   >
>   > The results show that incorporating the mask can lead to a performance improvement. In the final version, we will include these experimental results into the appendix.
>
> - **Question 2.2**: Have the authors ablated how simply switching out their mask would perform in other Vision-Mamba architectures?
>
>   > **Answer**: We would like to clarify that the proposed **plug-and-play mask is designed for attention variants**, such as vanilla self-attention, linear attention, and sparse attention—not for all architectures.
>   >
>   > Most existing Vision-Mamba works (e.g., VMamba) are based on Mamba1, which cannot be reformulated into an attention variant like Mamba2. Thus, our mask cannot be simply integrated into these architectures.
>   >
>   > As stated in Section 3, Mamba2 can be reformulated as the **structured masked linear attention**:
>   > $$
>   > Mamba2:{\rm SMA}(x)=(QK^\top \odot L^{1D})x,
>   > $$
>   >
>   > Thus, our polyline path mask $L^{2D}$ can be directly used to replace $L^{1D}$, while maintaining a complexity of $\mathcal{O}(N)$:
>   > $$
>   > {\rm PPMLA}(x)=((QK^\top)\odot L^{2D})V.
>   > $$
>   > However, Mamba2-based vision models are still relatively rare. Therefore, we have not evaluated our mask on them. We consider this a valuable direction for future work as the Mamba2 ecosystem matures.
>
> - **Question 3**: It would be interesting to add throughput to Table 4 to have a comparison of benefit vs. throughput decrease!
>   > **Answer**: Thanks for the suggestion. We have added **throughput** in Table 4. The results show that all models using learnable masks tend to have relatively low throughput.
>   >
>   > \begin{array}{lccc}\hline \text{Model}&\text{Throughput (imgs/s)}&\text{Top-1 Acc.} (\\%) &\text{SS mIoU} (\\%) \newline\hline \text{Baseline (w/o mask)}&1779&82.28&47.78\newline \text{+ RMT Decay Mask}&1650&82.35&48.01\newline \text{+ Cross Scan Mask}&1100& 82.44&48.14\newline \text{+ Hilbert Scan Mask}&1091&82.41&48.19\newline \text{+ V2H Polyline Path Mask}&1203&82.44 &48.57\newline \text{+ 2D Polyline Path Mask}&1034&\bf{82.60}&\bf{48.73} \newline \hline\end{array}
>
> - **Question 4**: What is the difference between your scanning approach to 2DMamba?
>   > **Answer**: While our scanning strategy is similar to 2DMamba, our core contribution lies in **integrating Mamba2’s structured mask mechanism into ViTs**. The detailed distinctions include:
>   >
>   > **1) Plug-and-play Flexibility.** We would like to reiterate that 2DMamba lacks the plug-and-play flexibility as ours. Specifically, Mamba1-based 2DMamba uses a diagonal matrix $A$ as the decay parameter, whereas our method follows Mamba2 and adopts a scalar $a$ instead. **This distinction is crucial**, as only a scalar decay factor allows for the derivation of the polyline path mask and the efficient algorithm, which in turn supports plug-and-play usage.
>   >
>   > **2) Independent Horizontal and Vertical Decay Factors.** Our method uses **separate** horizontal decay factor $\alpha_{i,j}$ and vertical decay factor $\beta_{i,j}$—to capture directional information separately. In contrast, 2DMamba uses **a shared decay parameter $A$**. Ablation study in Appendix B.2, Table 2 (Line 6 vs. Line 7) shows that a shared decay factor leads to a **0.23%** drop on ImageNet-1K.
>   >
>   > **3) V2H+H2V Scanning.** Unlike 2DMamba’s one-way horizontal-to-vertical (H2V) scanning, our method combines **V2H and H2V scanning** to ensure **symmetric decay** between tokens, i.e., the decay weight from token $x_{i,j}$ to $x_{k,l}$ equals its reverse.  Table 4 (Lines 4 vs. Line 5) shows that this improves accuracy by **0.16%** on ImageNet-1K.
>   >
>   > We will revise the appendix to clarify these distinctions. Lastly, since 2DMamba was released very recently and our scanning strategy was developed independently, we consider this a **concurrent work**.
>
> - **Question 5 (*Minor*)**: Table 4 and Figure 6 have a slight mismatch that could be improved.
>   > **Answer**: **Hilbert-Mask results** have been added to Table 4 (please see our response to Question 3). We will revise Table 4 and Figure 6 in the final version as suggested.
>
> - **Question 6**: More detailed limitations in the appendix could be moved to the main paper if space permits in the final version.
>   > **Answer**: Thank you for the suggestion. We will move detailed limitations to the main paper as suggested.
>
> We hope our responses have addressed your questions satisfactorily. The revision, incorporating your suggestions, has indeed made the paper more comprehensive. We’d be grateful for a higher rating if you feel it’s warranted.

---

> > ### Comment · Reviewer_gCJq · 2025-08-04
> > **Thanks for the detailed response!**
> >
> > I'd like to thank the authors for their detailed response to all my questions, and the additional insights and clarifications they have provided.
> >
> > I have also read through the other reviewers' comments, and respective rebuttals.
> > $\rightarrow$ Apart from my own questions, I very much agree with reviewer *8Ffr*'s suggestion to move Figure 5 from the appendix to the main paper -- even though the structure is not the main contribution and follows RMT (which should be mentioned in the caption), it simply makes the interpretation of the structure much easier for the reader!
> >
> > If space permits, moving some of the visualization of the decay factors in Figure 7 into the main paper (or at least referring to the appropriate section in the appendix)  would be great was well, as it's quite insightful to see the 'expectation' align with what's learnt.
> >
> > ---
> > On more note: The limited GPU memory of your resources does in fact not prevent you from training with bigger batch sizes, as you can always use *gradient accumulation* (i.e. only perform the actual 'update' after 2 backward passes -- the gradients get automatically summed up);
> > $\rightarrow$ This will provide you with the possibility to train on 2048 or even larger ones.
> >
> > ---
> > ---
> > **All in all, my concerns and questions have been sufficiently addressed, and I've raised my score accordingly.**

---

> > > ### Author Response · Authors · 2025-08-04
> > >
> > > We sincerely thank you for the constructive feedback and for recognizing our responses. As suggested, we will move Figure 5 from the appendix to the main paper, and ensure that Figure 7 is clearly referenced in the main paper.
> > >
> > > We are especially grateful for the valuable suggestion on gradient accumulation. And we will explore this approach in future experiments to enable training with larger batch sizes.

---

### Official Review · Reviewer_xW4s · 2025-07-02

**Clarity:** 3
**Significance:** 2
**Originality:** 2
**Rating:** 5
**Confidence:** 3

**Summary:**

This submission presents a method that builds upon Mamba2 for vision. It defines polyline "scanning paths" based on the L-shaped manhattan path between pairs of image/spatial locations. In the "dual" attention formulation this is an attention mask based on the product of decay factors along the polyline. There is an efficient way to compute this mask based on decomposing it into 1D vertical and horizontal scanning. The forward pass is computed using the dual form as structured mask attention, as part of an overall backbone architecture with 4 stages with two attention different variants using this mask. It's tested and compared to other architectures on ImageNet-1K, COCO, and ADE20K.

**Questions:**

- **Q1**: Is [36] intended to be included as one of the SSM-based comparison methods in the citation around line 261 in Section 5?
- **Q2**: Around line 215 in Section 4.2 it is stated that the lower triangular parts of $A^k$ and $B^j$ are 1-semiseparable. Is the argument (however brief) as to *why* matrices constructed this way are 1-semiseparable stated explicitly in the paper? I did not see it.
- **Q3**: Footnote $^2$ on page 4 states "We apply the ReLU and exponential operator after the MLP layers to ensure $\alpha_{i,j}, \beta_{i,j} \in [0, 1]$. What's the formula that applies these operators to ensure this? Do these values correspond to the `dt_alpha` and `dt_beta` on line 170 of https://anonymous.4open.science/r/PPMA-3948/classification/models/PPMA.py?
- **Q4**: Which result does https://anonymous.4open.science/r/PPMA-3948/training_logs/classification/ppma_small_log.txt correspond to? Is the Top-1 test accuracy for this run reported as 84.1% or 84.2%?

**Ethical Concerns:**

["NO or VERY MINOR ethics concerns only"]

**Final Justification:**

The answers to Q1-Q3 can be addressed by changes to the camera-ready that the authors describe. Though the statement connected it to Mamba2's mathematical results might benefit from a more mathematically formal description than "we adopt the same 1D scanning strategy as in Mamba2" to allow checking this result more carefully.

For Q4: The authors are correct that this is a standard technique in reporting eval results in the literature now. I have seen it in other contexts too. I do not like that this is the standard, but it is, so it makes sense not to count it against my score, to be consistent with the precedent from esteemed researchers & senior reviewers like our ACs & PCs.

So I raise to "5: Accept"

**Limitations:**

Yes

**Quality:**

4

**Strengths And Weaknesses:**

There's a number of works that consider masks or other means to impose the spatial prior on vision transformers and state space models. This submission presents experiments on a good selection of comparison methods that also include spatial relationships, such as [6, 40, 44]. (Plus "Vicinity Vision Transformer" by Sun et. al. T-PAMI 2023 also considers Manhattan distance, on the attention/transformer side.)

### Strengths

1. Experiments consider a variety of image tasks.

Notably, this includes both image-level labels and dense prediction (instance segmentation with COCO and semantic segmentation with ADE20K). We might expect that the role of local vs global information might be different between these ("MambaOut: Do We Really Need Mamba for Vision?" by Yu and Wang, CVPR 2025), and both are empirically tested.

Comparisons include backbone size and computation cost to make it clear what confounding tradeoffs might be present in those metrics between different architectures.

2. Gives a computation complexity analysis of both computing the mask and the overall attention.

3. Structure in PPMA intuitively gives higher weight based on locality.

4. Very well-organized code, including training logs

### Weaknesses

5. Presents only a structured attention form of the method, not the state space model form. (Unless I have missed this.) The motivation of the paper emphasizes how this method is constructed to build on Mamba2.

6. Overall accuracy differences on a lot of the comparisons are quite small. (0.1-0.2% in Table 1) While tending to also require a bit more computation (FLOPs) than the base RMT architectures.

Generally, this seems like a notably solid and well-executed paper with middle-range potential impact and originality.

---

> ### Author Rebuttal · Authors · 2025-07-30
>
> We are honored to receive the reviewer's recognition of our work as **“a notably solid and well-executed paper”**, and the acknowledgment of its **"potential impact and originality"**. Our work is inspired by Mamba2 and introduces a new structured masked attention mechanism that enhances the state space duality (SSD) framework in Mamba2. Please see below for our point-by-point responses to your comments.
>
> - **Reference Suggestion:** Plus "**Vicinity Vision Transformer**" by Sun et al. T-PAMI 2023 also considers Manhattan distance, on the attention/transformer side.
>
>   > **Response**: We will include a discussion of this work in the related work section.
>
> - **Weaknesses 1**: Presents only a structured attention form of the method, not the state space model form. (Unless I have missed this.) The motivation of the paper emphasizes how this method is constructed to build on Mamba2.
>
>   > **Response**: We acknowledge that our method, PPMA, is only presented in a structured masked attention mechanism, without any traditional state space model form. However, we want to emphasize that the close relationship between state space models and structured masked attention has been explicitly discussed in Mamba2 [4]. As detailed in Section 5.3 and Figure 4 of Mamba2 [4], **"state space models with scalar-identity structure on the $A$ matrices, and structured masked attention with 1-semiseparable structure on its $L$ mask – are duals of each other with the exact same linear and quadratic forms."**  This duality enables more flexible methodological design, and the proposed PPMA is built upon this foundation.
>   >
>   > **Notably, the proposed method is not directly constructed by human intuition alone, but is naturally derived through refinements within the Mamba2 framework**. The proposed polyline path mask $L^{2D}$ is a 2D adaptation of Mamba2’s $L^{1D}$, maintaining the same approach for generating decay factors (an MLP followed by ReLU and an exponential operator). The primary difference is that we replace the original 1D scanning strategy in Mamba2 with our 2D polyline scanning strategy.  Moreover, in terms of the overall architecture, the proposed method remains consistent with Mamba2:
>   > $$
>   > Mamba2: {\rm{SMA}}\left( x \right)  = \left( {Q} {K^{\top}} \odot {L}^{1D} \right) {x} ,
>   > $$
>   >
>   > $$
>   > Ours: {\rm{PPMVA}}\left( {{x}} \right)
>   > 			= \left( {{\rm softmax}(Q{K^ \top }) \odot {L^{2D}}} \right)V.
>   > $$
>   >
>   > Therefore, we claim that our method is based on Mamba2. We will clarify this more clearly in the revised manuscript.
>   >
>   > ------
>   >
>   > [4] Dao T, Gu A. Transformers are ssms: Generalized models and efficient algorithms through structured state space duality[J]. arXiv preprint arXiv:2405.21060, 2024.
>
>
>
> - **Weaknesses 2**: Overall accuracy differences on a lot of the comparisons are quite small. (0.1-0.2% in Table 1) While tending to also require a bit more computation (FLOPs) than the base RMT architectures.
>
>   > **Response**: Thank you for your feedback. We would like to make the following clarifications:
>   >
>   > **1) SOTA baseline makes further improvements quite challenging.** We selected the state-of-the-art RMT model as our baseline without incorporating any incremental tricks.  This ensures the fair comparison with the RMT method on the fundamental modules. However, RMT already achieves very high performance on ImageNet1k, making further improvements quite challenging.
>   >
>   > **2) The proposed PPMA achieves more significant improvements on fine-grained tasks.** As shown in Appendix Figure 7, the polyline path mask effectively suppresses interference from distant and irrelevant tokens, leading to more semantically accurate masked attention maps. This makes PPMA especially effective for dense prediction tasks such as semantic segmentation. For example, PPMA-S models outperform RMT-S by **1.3% SS mIoU** and **2.3% MS mIoU** on the ADE20K dataset, respectively. To further verify this, we additionally conducted experiments on the **Cityscapes** semantic segmentation task (results shown in the table below). Results show that PPMA-T outperforms RMT-T significantly by **0.8% SS mIoU** and **0.6% MS mIoU**, respectively.
>   >
>   > \begin{array}{lccc}\hline \text{Backbone}&\text{Input Size}&\text{SS mIoU} (\\%) &\text{MS mIoU} (\\%) \newline\hline \text{RMT-T}  & 1024\times1024 & 81.9 & 82.9 \newline \text{PPMA-T} & 1024\times1024 & \bf{82.7} & \bf{83.5} \newline \text{RMT-S}  & 1024\times1024 & 83.3 & 83.9 \newline \text{PPMA-S} & 1024\times1024 & \bf{83.7} & \bf{84.0} \newline \text{RMT-B}  & 1024\times1024 & 83.6 & 84.1 \newline \text{PPMA-B} & 1024\times1024 & \bf{83.9} & \bf{84.3} \newline \hline\end{array}
>   >
>   > **3) PPMA can degenerate into RMT.** Importantly, we would like to emphasize that **RMT can be viewed as a special case of PPMA**, where all learnable decay factors are fixed constants. Therefore, **our PPMA model is theoretically at least as expressive as RMT**. The learnable decay factors, while offering greater performance potential, inevitably lead to an increase in FLOPs.
>
>
>
> - **Question 1:** Is [36] intended to be included as one of the SSM-based comparison methods in the citation around line 261 in Section 5?
>
>   > **Answer**: **EffNet-B3 [36] is a CNN-based model**, not an SSM-based method. We included it for comparison following VMamba.
>   >
>   > ------
>   >
>   > [36] Tan M, Le Q. Efficientnet: Rethinking model scaling for convolutional neural networks[C]//International conference on machine learning. PMLR, 2019: 6105-6114.
>
>
>
> - **Question 2**: Why the lower triangular parts of $A_k$ and $B_j$ are 1-semiseparable?
>
>   > **Answer**: As stated in line 255 in Section 4.2, **matrices $A_k$ and $B_j$ correspond to 1D horizontal scanning along each row and 1D vertical scanning along each column, respectively.** Particularly, we adopt **the same 1D scanning strategy as in Mamba2** to construct matrices $A_k$ and $B_j$. This implies that the lower triangular parts of $A_k$ and $B_j$ are 1-semiseparable, since Mamba2 [4] has shown that a matrix constructed in this way is 1-semiseparable.
>   >
>   > To avoid potential confusion, we will revise the text in Section 4.2 to provide a clearer explanation.
>
>
>
> - **Question 3**: Footnote 2 on page 4 states "We apply the ReLU and exponential operator after the MLP layers to ensure αi,j,βi,j∈[0,1]. What's the formula that applies these operators to ensure this? Do these values correspond to the `dt_alpha` and `dt_beta` on line 170 of PPMA.py?
>
>   > **Answer**: Yes, thank you for carefully reviewing our code. These values indeed correspond to the `dt_alpha` and `dt_beta` on line 170 of the code. The precise formula is:
>   > $$
>   > \alpha_{i,j}= {\rm exp}(-A \times {\rm SoftPlus}({\rm MLP}(x_{i,j}))),
>   > $$
>   > where scalar $\alpha_{i,j} \in [0,1]$, $x_{i,j} \in \mathbb{R}^{1 \times C}$ denotes the input token, $\rm SoftPlus(\cdot)$ is a smooth alternative to $\rm ReLU(\cdot)$  to ensure ${\rm SoftPlus}({\rm MLP}(x_{i,j}))>0$, and $A$ is a positive parameter. Decay factor $\beta _{i,j} \in [0,1]$ is computed using the same formula. Here, this formulation follows the decay factor computation approach in Mamba2 [4] and corresponds exactly to the code.
>   >
>   > ------
>   >
>   > [4] Dao T, Gu A. Transformers are ssms: Generalized models and efficient algorithms through structured state space duality[J]. arXiv preprint arXiv:2405.21060, 2024.
>
> - **Question 4**: Which result does `ppma_small_log.txt` correspond to? Is the Top-1 test accuracy for this run reported as 84.1% or 84.2%?
>
>   > **Answer**: It is **84.2%**, achieved at **epoch 276** (`"test_ema_acc1": 84.186`).  Specifically, we follow the commonly adopted evaluation protocol on ImageNet-1K, consistent with methods such as RMT, Spatial-Mamba, and VMamba, by reporting the **best validation epoch** (rather than the final epoch) as the final result. Additionally, we adopt the same EMA (Exponential Moving Average) strategy as used in RMT to mitigate overfitting (please refer to Appendix B.2, Line 234).

---

> > ### Comment · Reviewer_xW4s · 2025-08-05
> >
> > **Q1:** Note that the original submission, around line 261 on Page 7, reads:
> >
> > "Comparison methods include advanced CNN-based [30, 29, 36], SSM-based [27, 44, 36, 42, 40], and Transformer-based backbones..." I only didn't understand why the citation to 36 was included in the second list among SSM-based methods. Thanks for explaining!
> >
> > **Q3:** I see, thanks! To make sure I clarify my understanding, in your rebuttal you mean that $\alpha_{i,j} = \exp(-A \times \mathrm{SoftPlus}(\mathrm{MLP}(x_{i,j})))$ represents the following code (from line 172-80 of classification/models/PPMA.py)?
> >
> > ```
> > A = -torch.exp(self.A_log)              # (nheads), torch.Size([2])
> > dt_alpha = F.softplus(dt_alpha + self.dt_bias)      # (B, L, 2*nheads), torch.Size([12, 3136, 2*4])
> > dt_alpha = dt_alpha*A                              # (B, L, 2*nheads), torch.Size([12, 3136, 2*4])
> > dt_beta = F.softplus(dt_beta + self.dt_bias)      # (B, L, nheads), torch.Size([12, 3136, 4])
> > dt_beta = dt_beta*A                              # (B, L, nheads), torch.Size([12, 3136, 4])
> >
> > dt_alpha = dt_alpha.view(batch, height, width, self.nheads).contiguous()
> > dt_alpha = dt_alpha.permute(0, 1, 3, 2).contiguous()        # (B, height, nheads, width)
> > structed_mask_w = self.generate_structed_mask_1d(dt_alpha)  # (B, height, nheads, width, width)
> > ```
> >
> > I'm now also just trying to track down where the outermost $\mathrm{exp}$ comes in, perhaps I've missed it in the code or it corresponds to something else in the math.
> >
> > **Q4:** To make sure I understand, is the validation set that is used to select the epoch the same as the validation set used to compute the accuracy reported in Table 1?

---

> ### Comment · Area_Chair_KbXf · 2025-08-05
> **Feedback Needed - Your AC**
>
> Dear Reviewer xW4s,
>
> I notice that the authors have submitted a rebuttal. Could you please let me know if the rebuttal addresses your concerns? Your engagement is crucial to this work.
>
> Thanks for your contribution to our community.
>
> Your AC

---

> ### Author Response · Authors · 2025-08-06
>
> We’re very pleased to receive your follow-up questions and truly appreciate your continued interest in our work. Please see below for our point-by-point responses.
>
> - **Questions 1:** Mistaken citation [36] :
>
>   > **Answer**: Thank you very much for pointing this out. We sincerely apologize for the oversight — **the citation to [36] was mistakenly included in the list of SSM-based methods**. This has been corrected in the revised version.
>
>
>
> - **Questions 3:** Where the outermost $exp(\cdot)$ of $\alpha_{i,j}= {\rm exp}(-A \times {\rm SoftPlus}({\rm MLP}(x_{i,j})))$  comes in?.
>
>   > **Answer**: Thanks for your question — you are absolutely right. To be more precise, the code from lines 172–180 corresponds to the following formula:
>   > $$
>   > α_{i,j}=-A \times \text{SoftPlus}(\text{MLP}(x_{i,j})).
>   > $$
>   > **The exponential function actually appears later in the softmax operation** within the self-attention mechanism. This design helps reduce the computational cost, since addition is more efficient than multiplication in PyTorch. Specifically, it appears in the following code (from line 366–367 of `classification/models/PPMA.py`):
>   >
>   > ```
>   > qk_mat = (0.5 * torch.softmax(qk_mat + structed_mask, -1) +
>   >           0.5 * torch.softmax(qk_mat + structed_mask.permute(0, 1, 3, 2), -1))
>   > ```
>   >
>   > Here, we exploit the identity:
>   > $$
>   > \text{Softmax}(Q_iK^{\top}+M_i)=\frac{e^{Q_iK^{\top}+M_i}}{\sum\limits_{i=1}^{N}e^{Q_iK^{\top}+M_i}}=\frac{e^{Q_iK^{\top}}\odot e^{M_i}}{\sum\limits_{i=1}^{N}e^{Q_iK^{\top}} \odot e^{M_i}}
>   > $$
>   > where $Q_i \in \mathbb{R}^{1 \times C}, K \in \mathbb{R}^{N \times C}$,  $M_i \in \mathbb{R}^{1 \times N}$ denotes the proposed structured mask, $N$ and $C$ denote the sequence length and channel number, respectively.
>   >
>   > We apologize for not making this clearer in the original submission due to space constraints, and we will clarify this point  in the final version.
>
>
>
> - **Questions 4:** Is the validation set that is used to select the epoch the same as the validation set used to compute the accuracy reported in Table 1?
>
>   > **Answer**: Yes, that's correct. As indicated in the title of Table 1, *"Image classification performance on the ImageNet-1K **validation set**"*, the same validation set is used both for selecting the best epoch and for reporting the accuracy.
>   >
>   > This practice follows a widely adopted and standard protocol in experiments on ImageNet-1K. All compared methods (including RMT, BiFormer, Spatial-Mamba, and V-Mamba) use the same validation set for both model selection and evaluation, ensuring a fair comparison.
>
> We hope our answers have satisfactorily addressed your concerns. If you have any further questions, please don’t hesitate to let us know — we would be happy to continue the discussion!
>
> If you feel our clarifications have been helpful, we’d be sincerely grateful for a higher score. It would mean a lot to us！

---

### Official Review · Reviewer_8Ffr · 2025-07-03

**Clarity:** 3
**Significance:** 3
**Originality:** 3
**Rating:** 5
**Confidence:** 4

**Summary:**

The authors propose a novel method called ​​Polyline Path Masked Attention (PPMA)​​, which effectively integrates the structured masking concept from Mamba2 with the self-attention mechanism in Vision Transformers (ViT). Specifically, the paper first introduces an innovative ​​2D Polyline Path Scanning​​ strategy that better preserves the 2D adjacency relationships among image tokens. Building upon this, the authors derive a corresponding ​​polyline path mask​​ formulation. The experimental results demonstrate that the proposed PPMA model achieves state-of-the-art (SOTA) performance across various scales.

**Questions:**

See my  comments on the weakness.

**Ethical Concerns:**

["NO or VERY MINOR ethics concerns only"]

**Final Justification:**

The authors have addressed all my questions.

**Limitations:**

While the paper presents a well-motivated fusion of Mamba and ViT with thorough theoretical analysis and extensive experiments, several key questions remain unaddressed. The network architecture design (Figure 5) lacks justification for its hybrid use of PPMCCA and PPMVA Blocks, with no ablation study to validate this choice. The learned decay factors (α_ij and β_ij) in the polyline path mask are not analyzed for interpretability (e.g., correlations with image semantics or edges), missing an opportunity to strengthen the method’s transparency. Presentation issues arise, such as relegating Figure 5 to the appendix and an unclear description of attention stages in Section 4.4. Discrepancies in RMT benchmark results versus the original paper raise reproducibility concerns, while the absence of a larger PPMA-L model leaves open whether scaling could close the performance gap with RMT in tasks like COCO detection/segmentation. Addressing these would solidify the work’s rigor and practical insights.

**Quality:**

3

**Strengths And Weaknesses:**

Strengths：
1. The research motivation is well-justified, as combining the strengths of Mamba and ViT represents a promising direction.
2. The paper provides theoretical analysis of the proposed polyline path masking and derives its computationally efficient algorithm, demonstrating solid foundational work.
3. Extensive experiments have been conducted across multiple vision tasks with comprehensive comparisons against various baseline methods.


Weaknesses:
1. In Figure 5, why is the network structure designed this way? Why not use only PPMCCA Blocks or only PPMVA Blocks? If this network architecture is optimally designed, could the paper provide corresponding analysis?
2. The decay factors (α_ij and β_ij) in the polyline path mask are learned through MLP. The paper could briefly discuss whether these learned factors exhibit any interpretable patterns, such as potential correlations with the semantic content or edge structures of the image.
3. It is recommended to relocate Figure 5 from the appendix to the main text to provide readers with more intuitive visual understanding.
4. Revise “where the first three stages employ Polyline Path Masked Attention” in Section 4.4 to “where the first three stages employ Polyline Path Masked Criss-Cross Attention”
5. Why do the RMT results in some tables differ from those reported in the original paper, while other models remain consistent with the results in the RMT paper?
6. I'm curious whether there is a PPMA-L model, because as the model size increases, the performance gap between PPMA performance and RMT is narrowing, such as the effect of object detection and instance segmentation in COCO val2017.

---

> ### Author Rebuttal · Authors · 2025-07-30
>
> We sincerely thank the reviewer for acknowledging our **research motivation, theoretical analysis with efficient algorithm design, and comprehensive experiments**. Please see below for our point-by-point responses to your comments.
>
> - **Question 1 (Weaknesses 1)**: In Figure 5, why is the network structure designed this way?
>
>   > **Answer**: **The utilized network structure in Figure 5 strictly follows the structure design of previous SOTA work in the same category, RMT ([8] in the manuscript),  for a fair comparison.**
>   >
>   > This hybrid design balances computational efficiency with performance. In the early stages of the network, the token sequence length is large. Employing criss-cross attention (PPMCCA) in these stages significantly reduces computational complexity, since it is a sparse attention variant. In the final stage, the token sequence is much shorter, so using full vanilla self-attention (PPMVA) introduces only marginal computational overhead while enhancing the model’s ability to capture global dependencies. Such a design should be superior to using only PPMCCA or PPMVA, but whether it is the optimal design still requires further verification.
>   >
>   > In this paper, **the overall network architecture is not the focus of our research**. Therefore, to facilitate fair comparisons with previous methods, we chose this design and leave the exploration of better overall structures for future work.
>   >
>   > Following the reviewer’s suggestion, we will appropriately incorporate the aforementioned network structure analysis into the final version of the paper if it is accepted.
>
> - **Question 2 (Weaknesses 2)**: The paper could briefly discuss whether these learned factors exhibit any interpretable patterns, such as potential correlations with the semantic content or edge structures of the image.
>
>   > **Answer**: We appreciate the reviewer's insightful comment. Actually, we have already provided a discussion on our insight about decay factors in **Appendix B.6 and C.1**, through detailed experimental observations and theoretical analysis. Our conclusions are fully consistent with the reviewer's speculation: **these factors do exhibit correlations with the semantic content or edge structures of the image.**
>   >
>   > **1) Interpretable Visualization Results.** The supervised learning process encourages the horizontal decay factors ($α_{ij}$) and vertical decay factors ($β_{ij}$) to focus on horizontal and vertical edge information, respectively. As shown in Figure 7 of Appendix B.6 (particularly the cup example in the 8th row), we observe that the learned decay factors exhibit interpretable patterns that correspond well to image structures such as edges and semantic boundaries.
>   >
>   > **2) Theoretical Insight: Connection to the Local Smoothness Prior in images.**  As further analyzed in Appendix C.1, an ideal polyline path mask should align well with the *local smoothness prior* commonly assumed in natural images. This implies that:
>   >
>   > - Within homogeneous regions that share the same semantics, the decay factors ($α_{ij}$ and $β_{ij}$) should be large (closer to 1), maintaining continuity in the polyline paths.
>   > - At semantic boundaries or edge structures, these decay factors should be small (closer to 0), allowing the mask to appropriately capture discontinuities.
>   >
>   > As suggested by the reviewers, we will include a brief discussion on the interpretability of decay factors in the final version of the paper, with appropriate references to the relevant experimental results and analyses in the Appendix.
>
> - **Question 3 (Weaknesses 3)**: It is recommended to relocate Figure 5 from the appendix to the main text.
>
>   > **Answer**: We will relocate Figure 5 (or a simplified version) to the final version if it is accepted.
>
> - **Question 4 (Weaknesses 4)**: Revise “where the first three stages employ Polyline Path Masked Attention” in Section 4.4 to “where the first three stages employ Polyline Path Masked Criss-Cross Attention”.
>
>   > **Answer**: Thank you for pointing out the issue.  We have revised it as suggested.
>
> - **Question 5 (Weaknesses 5)**: **Why do the RMT results in some tables differ from those reported in the original paper?**
>
>   > **Answer**: As the most important competing counterpart of our method, **we reproduced RMT results under the same experimental settings as our method** for a fair comparison.  Specifically, we trained both our models (PPMA) and the RMT baselines using a batch size of **1024**, due to hardware constraints—we only have access to a single 8-GPU (RTX 4090 with 24GB memory) machine, which cannot accommodate larger batch sizes because of limited GPU memory.
>   >
>   > **The reason for the discrepancies between our reported RMT results and those in the original paper lies in the batch size used during training**. The model weights provided in their GitHub repository were trained with a batch size of **2048** using **16 A100 GPUs (80GB memory)**. It is worth noting that, possibly due to an oversight by the RMT authors, the RMT paper states that a batch size of 1024 was used on ImageNet-1K, which differs from the information disclosed in the GitHub repository.  In contrast, we have rigorously verified through multiple repeated experiments that the results we provide truly reflect the expected performance of RMT under this configuration (batch size = 1024). Batch size can affect  classification results by **0.1% to 0.2%**, which also affects downstream COCO detection results since they rely on ImageNet-1K pretrained weights.
>   >
>   > In conclusion, we are confident that **all experiments in our paper—including RMT baselines—were conducted under fair and reproducible settings**. To ensure full transparency, we released the complete training code, conda environments, training logs, and model checkpoints for both **PPMA and reproduced RMT baselines** in the anonymous codes (hosted on Google Drive) a month ago.
>   >
>   > We will explicitly clarify the experimental settings in the final version to prevent any potential confusion. Please see the response to the first question of reviewer gCJq for more detailed explanation.
>
> - **Question 6 (Weaknesses 6)**: **I'm curious whether there is a PPMA-L model, because as the model size increases, the performance gap between PPMA performance and RMT is narrowing, such as the effect of object detection and instance segmentation in COCO val2017.**
>
>   > **Answer**: We  fully acknowledge the reviewer's valuable point that evaluating PPMA-L is indeed crucial for validating our method's performance on large-scale models. However, due to **limited computational resources** in our lab (equipped with only 8×RTX 4090 GPUs), we were unable to conduct PPMA-L experiments under standard benchmark configurations. Therefore, to maintain scientific rigor, we opted not to include PPMA-L results in the current manuscript.
>   >
>   > That being said, **we have in fact conducted a series of PPMA-L experiments on smaller-scale datasets.** Specifically, we have conducted preliminary comparisons between **PPMA-Large (PPMA-L)** and **RMT-Large (RMT-L)** (previous SOTA method) on the **ImageNet100** dataset (using a smaller batch size of 512). The results (summarized in the table below) show that **PPMA-L outperforms RMT-L by 0.3%** top-1 accuracy, suggesting that the performance gap does not vanish at larger scales and that PPMA’s effectiveness generalizes across model sizes.
>   >
>   > \begin{array}{lccc}\hline \text{Model}&\text{Param. (M)}&\text{FLOPs (G)}&\text{Top-1 Acc.}(\\%) \newline\hline \text{RMT-L}  & 94.5 & 18.2 & 88.9 \newline \text{PPMA-L} & 94.6 & 18.6 & \mathbf{89.2} \newline \hline\end{array}
>   >
>   > In the final version, we will incorporate these additional experimental results into the appendix.

---

### Note · Authors · 2025-08-12

Dear Associate Chair and Reviewers,

We sincerely thank the AC and reviewers for their time, effort, and constructive suggestions. We will further revise the manuscript according to the valuable feedback. We have addressed each concern in detail, supported by extensive experiments, which can be summarized as follows:

**1) Regarding Experimental Results:**

- Reviewer 8Ffr’s Weaknesses 5 and Reviewer gCJq’s Question 1 concern the **the lower RMT reproduction results**.  We clear clarified that the **RMT reproduction issue arises from differences in batch size, which are necessary for a fair comparison.**  All the experiments in our paper were conducted under fair and reproducible settings. For more details, please refer to our response to Reviewer gCJq’s Question 1.

- We conducted extensive experiments to further validate the effectiveness of our method, including **classification (PPMA-Large) on ImageNet100**, **semantic segmentation on Cityscapes**, **object detection on COCO** with the 3× MS schedule, and **ablation studies on Deit baseline with and without PPMA**. Please refer to our response to Reviewer 8Ffr’s Weaknesses 6, Reviewer xW4s’s Weaknesses 2, and Reviewer gCJq’s Question 2.

- Reviewer gCJq’s Questions 3 and 5 offer constructive suggestions regarding the presentation of the experimental results. Accordingly, we have added **throughput results** and **Hilbert-Mask results** in Table 4.

**2) Regarding Methodology:**

- We clarified **the interpretability of the learned decay factors** (Reviewer 8Ffr’s Question 2) and provided its precise formula (Reviewer xW4s’s Question 3).

- Reviewer 8Ffr’s Question 1 concerns our network architecture design, which we clarified strictly follows the RMT baseline for fair comparison.

- Reviewer xW4s’s Question 2 and Reviewer SWqC's Questions concern the connection between our method and Mamba2, which we have clearly addressed in our response.

Additionally, we would like to reiterate our core contributions:

- The proposed polyline scanning strategy **fundamentally addresses the issue of destroyed 2D image structure**.
- Our work is **the first to integrate Mamba2’s structured mask mechanism into ViTs**.
- We designed **an elegant and efficient algorithm for the proposed mask** with rigorous theoretical guarantees.
- We conducted **extensive experiments** on mainstream high-level vision tasks, achieving SOTA performance among the latest Mamba-based and Transformer-based backbones.

Best regards,
Authors

---

### Decision · Program_Chairs · 2025-09-17

**Decision:**

Accept (spotlight)

**Comment:**

All the reviewers unanimously vote for acceptance. After checking the review, the rebuttal, the manuscript, and the discussion, the AC agrees with this assessment.